# Frontal polymerization-triggered simultaneous ring-opening metathesis polymerization and cross metathesis affords anisotropic macroporous dicyclopentadiene cellulose nanocrystal foam

Jinsu Park [1] & Seung-Yeop Kwak [1,2,3 ✉]

Multifunctionality and effectiveness of macroporous solid foams in extreme environments have captivated the attention of both academia and industries. The most recent rapid, energy-efficient strategy to manufacture solid foams with directionality is the frontal polymerization (FP) of dicyclopentadiene (DCPD). However, there still remains the need for a time efficient one-pot approach to induce anisotropic macroporosity in DCPD foams. Here we show a rapid production of cellular solids by frontally polymerizing a mixture of DCPD monomer and allyl-functionalized cellulose nanocrystals (ACs). Our results demonstrate a clear correlation between increasing % allylation and AC wt%, and the formed pore architectures. Especially, we show enhanced front velocity ($v_f$) and reduced reaction initiation time ($t_{init}$) by introducing an optimal amount of 2 wt% AC. Conclusively, the small- and wide-angle X-ray scattering (SAXS, WAXS) analyses reveal that the incorporation of 2 wt% AC affects the crystal structure of FP-mediated DCPD/AC foams and enhances their oxidation resistance.

[1] Department of Materials Science and Engineering, Seoul National University, 1 Gwanak-ro, Gwanak-gu, Seoul 08826, South Korea. [2] Research Institute of Advanced Materials (RIAM), Seoul National University, 1 Gwanak-ro, Gwanak-gu, Seoul 08826, South Korea. [3] Institute of Engineering Research, Seoul National University, 1 Gwanak-ro, Gwanak-gu, Seoul 08826, South Korea. ✉email: sykwak@snu.ac.kr

Functional cellular solids or polymeric solid foams are developed to suit the needs of specialized applications such as automotive, aerospace, electronics, and biomedical, where materials perform in extreme environments. Their lightweight, excellent thermal insulation, shock absorbance, and abilities to transport or store energy, gas, or fluids, and so on have been of particular interests in both academia and industries[1–6]. The customizable nature of these materials' properties is greatly affected by the tailored structures[7]. Therefore, advanced design strategies can elucidate the unfathomed structure-property relationships that are helpful for building forthcoming multifunctional materials. Recently, biomimetic cellular construction in foams has been a popular strategy to greatly enhance functionality of synthetic materials by recreating naturally occurring configurations such as anisotropy and honeycomb-like orientations[8–10]. Inspired by nature's structural complexity and topologies, more recent works have explored different foaming processes such as 3D printing and freeze casting to better control foam architectures and to realize the followed unique characteristics such as improved thermal and fire retardancy, and strength-to-weight ratio, respectively[11,12]. However, the requirement for additional preparation of 3D printable inks, specialized equipment, posttreatment procedures, and high energy and large capital inputs still limit the efficiency of foam manufacturing processes[13,14].

An emerging technique for producing porous materials that overcomes the forementioned challenges is frontal polymerization (FP). FP is a mode of converting monomer into polymer during an exothermic, self-sustained bulk polymerization. Initiated from a localized reaction zone, a propagating front autonomously cures the monomer by the coupling of thermal diffusion and Arrhenius reaction kinetics[15]. Therefore, the followed efficiencies in time, cost, and energy rose the desirability of FP in the manufacturing of thermoset polymers and composite materials. Its transition to up-scaling, however, was limited due to the lack of control over the pot life at room temperature (RT) by the Arrhenius kinetics. In efforts to alleviate this challenge, A. Mariani et al.[16] and I. D. Robertson et al.[17] reported the use of different Grubbs' catalyst (first generation, GC1; second generation, GC2)–inhibitor combinations (e.g., GC1–triphenylphosphine[16], GC2–phosphite[17]) that greatly reduced the bulk polymerization rate and extended the pot life during the frontal ring-opening metathesis polymerization (FROMP) of dicyclopentadiene (DCPD) monomer at RT. The versatility of FP reactions inspired for many advanced studies to investigate, for instance, the effects of different monomer-initiator mixtures[18] and reaction triggers[19] to the properties of resulting composites[20], and to the morphologies of the spontaneously created patterns realized by thermal instability[21,22]. Especially, cellular solids with apparent hollow topologies were effectively created via FP of DCPD. For example, M. Garg et al.[23] recently reported a production of porous or vascularized DCPD monolith via a synchronized ring-opening metathesis polymerization (ROMP) and depolymerization of DCPD and a poly(propylene carbonate) sacrificial layer, respectively. Moreover, D. M. Alzate-Sanchez et al.[24] studied the relationship between blowing agents, gelation time, and resin viscosity in the production of anisotropic channels within DCPD foams. Despite these cutting-edge results, a more homogeneous pore organization may be achieved by reducing the dependency of foam porosity to the number of sacrificial layers and the long delay time derived from precursor formulations. Within the foam manufacturing via FP of DCPD, a more precise and rapid means of controlling local pore morphology and size distribution are not yet realized. Thus, different strategies that tailor foam architectures are needed to produce more advanced, ubiquitous macroporous polymeric materials.

Interestingly, the intrinsic characteristics of foaming agents, or building blocks, such as the shapes and assembly also govern the resulting foam micro-/macrostructures and properties. For example, foams created with spherical $SnO_2$ nanoparticles (NPs)[25] and multi-walled carbon nanotubes[26] exhibited round and elongated pore microstructures that were accompanied by enhanced microwave absorption and electromagnetic shielding performance, respectively. Additionally, neat anisotropic packings of polymer nanofibers[27] and metal nanoplates[28] subsequently improved conductivity and ion transport of the produced foams, respectively. Amongst these materials, cellulose nanofibers (CNFs) and cellulose nanocrystals (CNCs) are unique building blocks that have gained an exponential amount of attention in composite and foam manufacturing for their innate anisotropic structure, structural robustness, renewability, and cost-efficiency[14,29,30]. Specifically, their structure-directing ability, enabled by the inherent cellulose morphologies, to form tree xylem-like honeycomb microstructures was observed in solid foams prepared by unidirectional freeze-casting[31–33], mold-casting[34], and ice-templating directional freeze-drying[35]. Despite these progresses, such organized morphological structures induced by cellulose are not yet realized through FP.

Currently, chemically induced phase separation (CIPS) and high internal phase emulsion (HIPE) polymerization are the two prevalent methods of producing DCPD foams[36]. We hypothesized that an FP-triggered simultaneous ROMP and cross metathesis (CM) of DCPD and CNCs, respectively, and the subsequent crosslinking rate differences would drive an effective phase separation and thus foam manufacturing. Herein, we report a rapid formulation of DCPD/CNC foams with homogeneous distributions of millimeter-scale anisotropic honeycomb-like pores, or millichannels, that are realized by the shape and surface chemistry of allyl-functionalized CNC (AC). Our results provide a direct correlation between AC wt% and the formed pore microstructure, and show the ability of ACs to alter the crystal structures of poly(DCPD) (pDCPD) that relatively inhibited oxidation. Based on these results, our work provides a facile method of tailoring foam morphology by using functionalized bioabundant CNCs and contributes to the development of simpler strategies that replicate biological pervasive vascular networks such as engineered tissues, organoids, synthetic gas exchange devices, heat exchangers, and flow batteries.

## Results and discussion

**Synthesis and characterization of AC.** In a standard production of porous pDCPD, a robust separation of the monomer and solvent phases largely contributes to the resulting foam structures and properties. Recently, the introduction of various metal NPs and lipase enabled the realization of distinctive foam morphologies and functionalities[36]. Similar phenomena were observed from FROMP-mediated foams by embedding secondary solvent[24] and polymeric microparticles[21] in the monomer layer. While different combinations of monomer-secondary phase systems still remain widely unexplored in DCPD FROMP, we were inspired to examine how the intrinsic morphology of rigid rod-like CNCs would affect the construction of the pDCPD monolith microstructure in situ. Therefore, our initial approach was to frontally polymerize a mixture of DCPD monomer and hydrophobically modified CNCs. CNCs were coated with tannic acid-decylamine (CTD) by following a method reported by Z. Hu et al.[37] (Supplementary Fig. 1). Compared to the μ-CT images of a neat DCPD solid, those of DCPD/1 wt% CTD exhibited formation of a few voids (Supplementary Fig. 2). Meanwhile, reactions of 2, 4 wt% CTD were not initiated presumably due to the poor

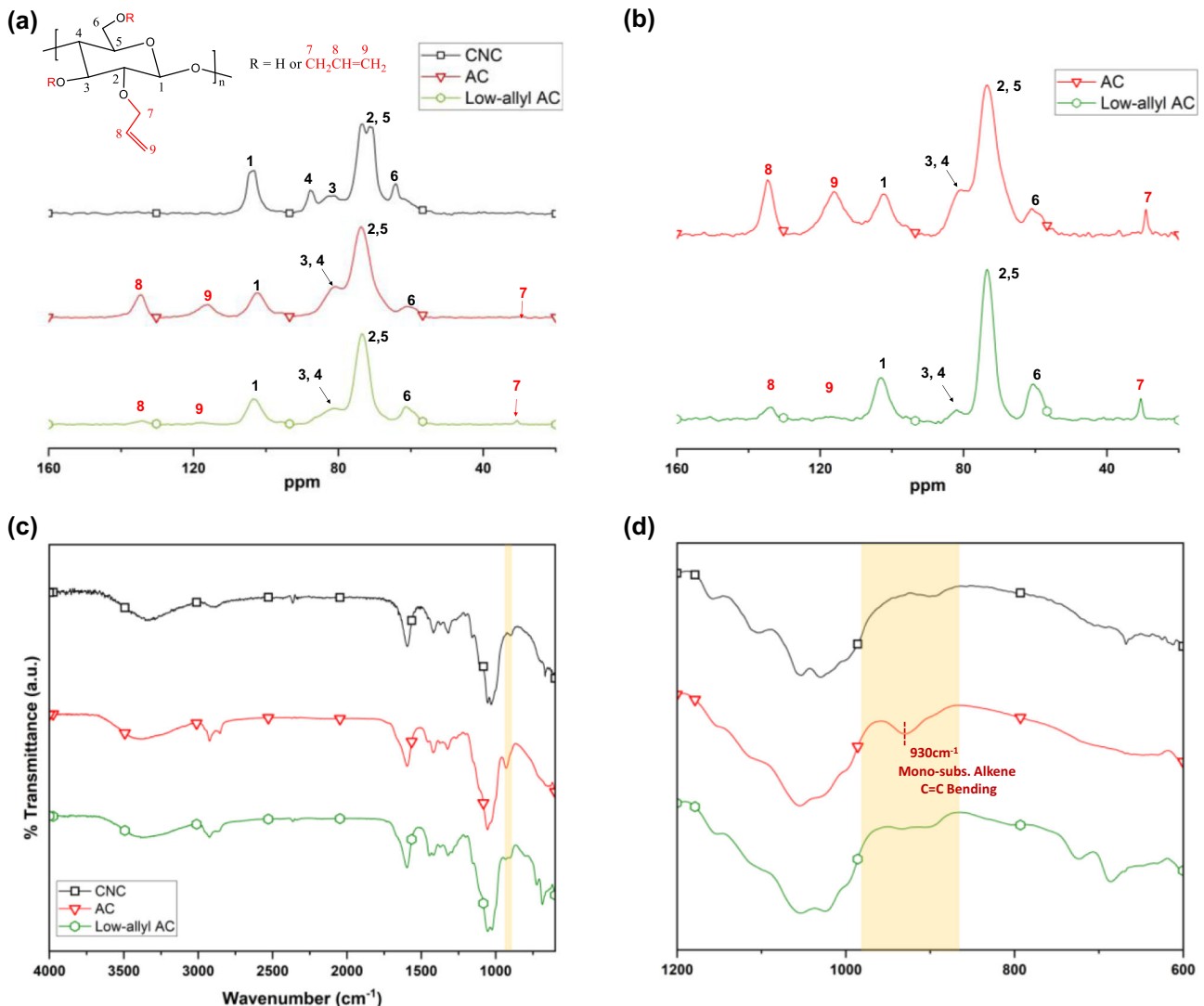

**Fig. 1 Preparation of allyl-functionalized cellulose nanocrystals of varying % allylation. a** Solid-state $^{13}$C CP-MAS NMR spectra of CNC, AC, and LAC. **b** $^{13}$C solid IG-NMR of AC and LAC, and the assigned allyl-group carbon peaks. **c** The normalized ATR-FTIR spectra of CNC, AC and LAC. **d** A representation of robust allylation of CNC by the strong monosubstituted alkene C=C bending at 930 cm$^{-1}$ in AC.

dispersity and hindrance in the reaction initiation induced by the large amount of tannic acid-decylamine moiety (Supplementary Fig. 3). These heterogeneous and isotropic pore organizations suggested that hydrophobic coating of CNC was insufficient to produce cellular solids.

In DCPD FROMP, GC2 is an important component that generates polymers by cleaving and forming C=C bonds between monomers. Leveraging this chemistry, we hypothesized that a more effective separation between the monomer and CNC phases may be achieved by introducing additional C=C moieties (R'CH$_2$CH=CH$_2$) to CNC (C$_6$H$_7$O$_5$), or by using AC (C$_6$H$_7$O$_5$R$_3$, R=H or CH$_2$CH=CH$_2$)[38]. Specifically, we hypothesized that the C=C moieties of AC would also undergo FP which then simultaneously triggers ROMP and CM of DCPD and AC, respectively. First, AC was synthesized via a one-step process using NaOH/urea aqueous solution and allyl-chloride[38]. Solid-state $^{13}$C cross-polarization magic angle spinning (CP-MAS) NMR spectra of CNC, AC, and low allyl-AC (LAC) (Fig. 1a, Supplementary Fig. 15, 17) and inverse-gated (IG) NMR spectra of AC and LAC were obtained (Fig. 1b, Supplementary Figs. 4, 5, 16, 18). For a quantitative analysis of the ratio between one AC AGU unit and its allyl-moieties,

IG-decoupling sequence was performed. By gating off the $^{13}$C spin over the acquisition time only, the $^{13}$C decoupling sequence reduces the Nuclear Overhauser Effect (NOE) enhancements of carbon atoms[39]. In the case of AC, the CP-MAS NMR analysis revealed carbon chemical shifts (δ) of its allyl groups (–CH$_2$CH=CH$_2$) at 134.5 ppm (C8), 116.0 ppm (C9), and 29.22 ppm (C7) which agreed with those from previous reports[38,40]. Carbon characteristic peaks of LAC were also observed at similar δ: 135.3 ppm (C8), 117.9 pm (C9), 30.69 ppm (C7) but exhibited weaker intensities. From both AC and LAC, the C7 peaks lacked clarity compared to C8 and C9 peaks presumably due to the newly built up NOE by several experimental parameters that disproportionally enhanced certain carbon signals and produced non-quantitative spectra[41]. The results obtained from the IG-NMR analyses of AC and LAC revealed allyl-group carbon peaks at similar δ values (δ$_{AC,C8}$ = 134.7 ppm, δ$_{AC,C9}$ = 116.1 ppm, and δ$_{AC,C7}$ = 29.15 ppm; δ$_{LAC,C8}$ = 133.6 ppm, δ$_{LAC,C9}$ = 117.2 ppm, and δ$_{LAC,C7}$ = 30.54 ppm). For the quantitative peak integration of C7–9 carbon peaks, C1 peak was set as a reference (1.00) (Supplementary Figs. 4, 5, 15–18). Using the typical method of calculating degree of substitution of cellulose[42,43], a complete

allylation or the sum of peak integration across C7–9 positions was assumed to be 3. Therefore, % allylation was obtained by

$$\% \text{ allylation} = \frac{(\text{sum of the obtained peak integrated values of C7, C8, C9})}{3}$$

Using this method, % allylation of AC and LAC were approximately 79% and 13%, respectively (Table 1). The significant difference in % allylation aligned well with the results exhibited in the normalized attenuated total reflectance-fourier-transform infrared spectroscopy (ATR-FTIR) spectra of AC and LAC, where the intensity of monosubstituted alkene bending at 930 cm$^{-1}$ was significantly weaker in LAC compared to AC (Fig. 1c, d)[38]. Collectively, these results confirmed a successful preparation of ACs with varying % allylation for FP.

**Synthesis and characterization of frontally polymerized DCPD/AC foams.** Both AC and LAC NPs were suspended in liquid DCPD monomer at four different wt%: 0.5, 1, 2, 4, prior to being thermally initiated for FP. It is widely understood that the NP dispersity greatly contributes to the characteristics of reaction products[44,45]. Therefore, NPs were agitated in liquid monomer layer at varying wt% using a probe ultrasonic processor and further processed in an ultrasonic bath until appropriately dispersed. Probe-ultrasonication was an essential step that prevented the agglomeration of AC and LAC in aqueous media, as confirmed in the transmission electron microscopy (TEM) analysis (Supplementary Fig. 6). Then, the monomer/NP mixtures were degassed and stirred in an N$_2$ environment to which the GC2 catalyst solution was added and further stirred briefly. Approximately 11,000 µL of each mixture was transferred into a 15 ×150 mm glass tube. Using a soldering iron, FP was initiated from the bottom of the glass tube from which bubbles started to form and white cylindrical DCPD/AC foams were subsequently produced in the direction of the propagating front (Supplementary Videos 1, 2, 3, 4).

With increasing AC wt%, the produced foams became paler and more porous with increasingly larger millichannels (Fig. 2a–c). Consequently, DCPD/AC foams exhibited decreasing densities (ρ) and volume fractions (ϕ) for increasing AC wt% (Supplementary Table 1). On contrary, DCPD/LAC foams remained less porous and relatively transparent even at higher LAC wt% which were translated into higher ρ and ϕ than those of DCPD/AC foams (Fig. 2d, Supplementary Table 2). µ-CT images of the formed cellular solids were obtained to confirm our observations and assess their morphological structures (Fig. 3). In the case of DCPD/AC foams, an increasing trend of pore

**Table 1 Quantitative allyl/AGU ratios and the calculated % allylation values of AC and LAC were obtained via $^{13}$C Inverse-gated Solid NMR (IG-NMR).**

|  | Quantitative Allyl/AGU ratio | % allylation |
| --- | --- | --- |
| Allyl-cellulose (AC) | 2.4:1 | 79 |
| Low allyl-AC (LAC) | 0.39:1 | 13 |

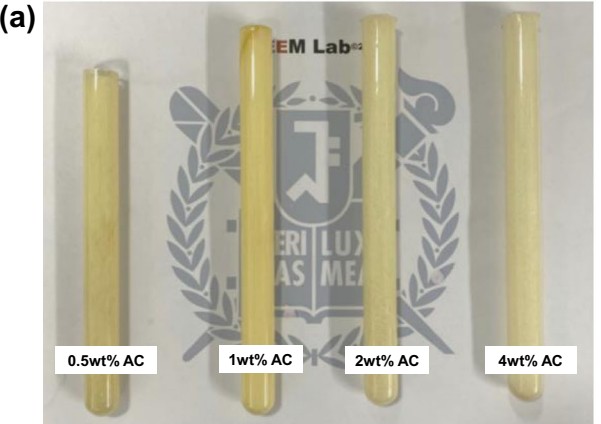

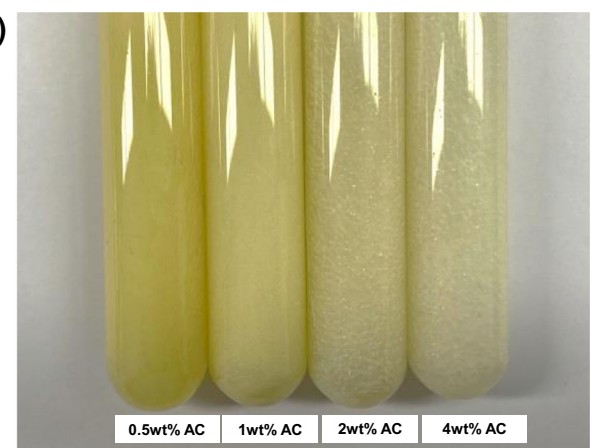

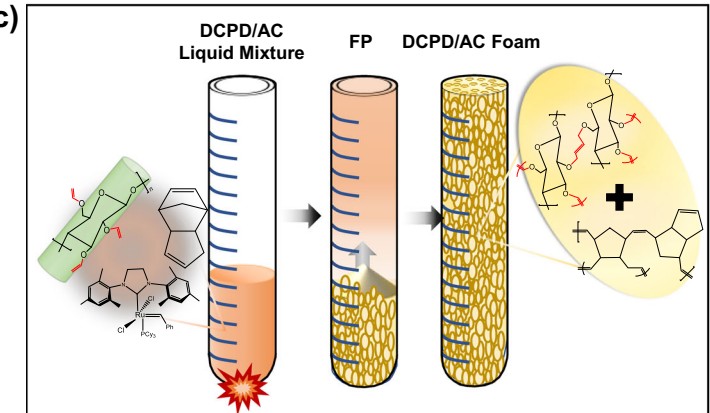

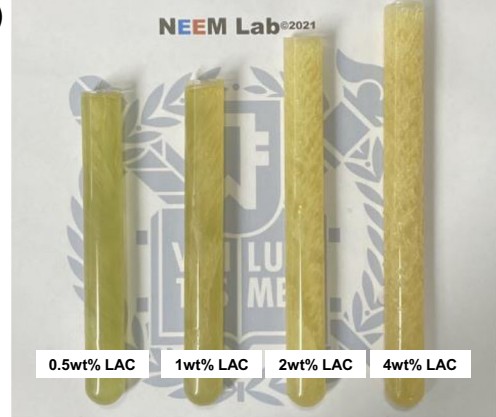

**Fig. 2 FP-mediated production of DCPD/AC, -LAC foams. a** Image of frontally polymerized DCPD/AC foams at varying AC wt%. **b** Pore sizes of DCPD/AC foams increased with increasing AC wt%. **c** A schematic showing the frontal manufacturing of DCPD/AC foams. **d** Image of DCPD/LAC foam that were produced under the same experimental set up as the DCPD/AC foams.

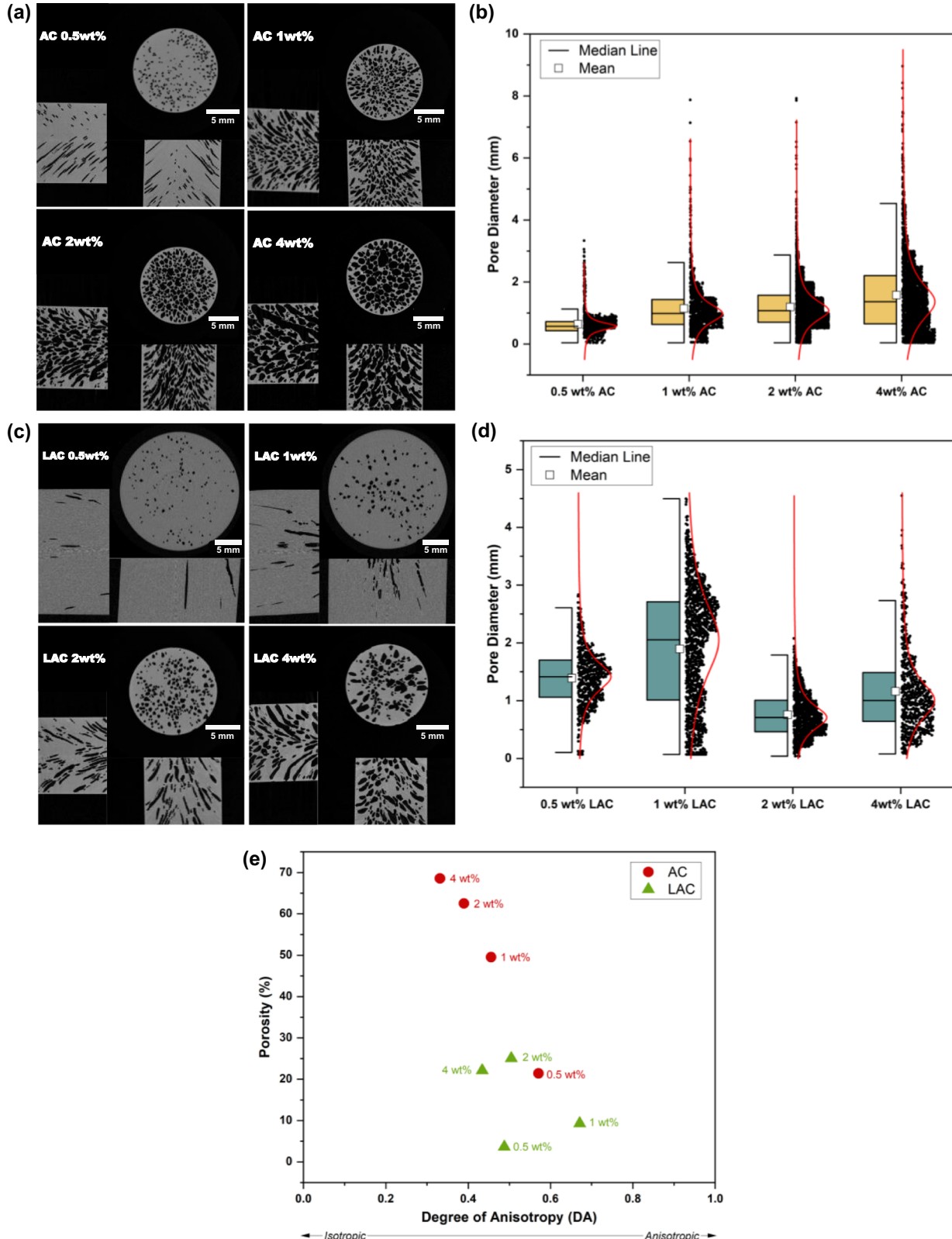

**Fig. 3 A directly proportional relationship between % allylation, AC wt%, and the produced pore sizes and organizations was observed. a** μ-CT images of DCPD/AC foams at varying AC wt%. **b** Box plot of the DCPD/0.5-4 wt% AC foam pore sizes and their homogenous distribution. **c** μ-CT images of DCPD/LAC foams at varying LAC wt%. **d** Box plot of DCPD/0.5-4 wt% LAC foam pore sizes and their heterogeneous distribution. **e** A plot showing the relationship between the % porosity and degree of anisotropy of DCPD/AC and -LAC foams.

**Table 2 Average diameters of DCPD/AC and DCPD/LAC foams, and their standard deviation values from three replicates are shown.**

| Avg. diameter (mm) | 0.5 wt% | 1 wt% | 2 wt% | 4 wt% |
|---|---|---|---|---|
| AC | 0.64 ± 0.4 | 1.1 ± 0.8 | 1.2 ± 0.7 | 1.6 ± 1 |
| LAC | 1.4 ± 0.5 | 1.9 ± 1 | 0.76 ± 0.4 | 1.2 ± 0.7 |

diameters was observed with increasing AC wt% and the pore measurements were in range of sub-to-inter-milipore class (Table 2)[46]. Moreover, the pore size distribution across the transverse plane became increasingly homogeneous until 2 wt% AC then slightly decreased at 4 wt% AC (Fig. 3a, b). The porosity (% porosity) and degree of anisotropy (DA) of DCPD/AC foams along the coronal planes were obtained via the Bruker CTAn software. As shown in Fig. 3e, the increases in % porosity and AC wt% were directly proportional while relatively maintaining the directional anisotropy within the range of DA that was previously reported of DCPD foams[24] ($DA_{0.5wt\%AC} = 0.57$; $DA_{1wt\%AC} = 0.46$; $DA_{2wt\%AC} = 0.39$; $DA_{4wt\%AC} = 0.33$). The slight decrease in DA was due to the interconnectivity of stochastically created anisotropic bubbles (Supplementary Fig. 7). On the other hand, LAC foams showed drastically contrasting microstructures despite being constructed under the same experimental procedures (Fig. 3c). Unlike the case of AC, DCPD/LAC foams exhibited rather random distributions of pore sizes, % porosity, and DA despite increasing LAC wt% (Figs. 3d, e). Based on these results, a clear correlation between % allylation, AC wt% and the resulting foam architecture was observed.

**AC morphology and its C=C functionalization tandem drives a physicochemical phase separation.** CIPS and HIPE are generally understood as a three-step biphasic polymerization system in which first, DCPD monomer is either dissolved in a solvent as a minor component or mixed with an immiscible liquid (80–99 vol%), then polymerized into a continuous phase[24,36,47]. Once the non-polymeric phases are removed, the formed macroporous structures are revealed. On the other hand, current study offers a one-pot strategy in which the separation of simultaneously polymerized majority (DCPD, 96–99.5 wt%) and minority (AC, 0.5-4 wt%) phases creates macroporous architectures in situ.

To support this claim, CP-MAS NMR analyses of the produced DCPD/AC foams were performed under the hypothesis that the C=C groups of DCPD and AC did not crosslink with each other during FP. Specifically, we aimed to examine any δ shifts or the C=C bond peak intensity changes at approximately 130 ppm (C8, C10) with increasing AC wt%. As shown in Supplementary Fig. 8, however, no noticeable differences between the CP-MAS NMR spectra of DCPD/AC foams and that of neat DCPD solid were observed. On the other hand, the X-ray diffraction (XRD) analysis of the same specimens revealed an increasing intensity of the cellulose (004) plane characteristic peak[37] at approximately $2\theta \approx 31.7°$ with increasing AC wt% (Supplementary Fig. 9). Collectively, these results indicated that the C=C moieties of AC and DCPD did not crosslink with each other during the one-pot FP-production of DCPD/AC foams.

The voids, or bubbles, within the foams were the second indicator that corroborated our hypothesis. According to previous studies, bubbles were generated by the evacuation of the individually dispersed droplets either during the formation of a new continuous internal phase[47], the elimination of the dissolved gas or water in monomer[48], and solvent boiling or thermal decomposition of initiators[49]. In the case of the current

study, however, the $N_2$-purged reaction environment and degradation of the reactants seemed less likely to have induced bubble formation. First, DCPD/0.5–1 wt% CTD solids, despite being produced in the same experimental conditions as those of DCPD/AC, -LAC foams, remained bubbleless. Moreover, as observed from the thermogravimetric analyses (TGA) (Supplementary Fig. 10), the decomposition temperatures of AC and LAC (~260 °C) were significantly higher than the maximum reaction temperatures ($T_{Max}$) obtained from this study (~150 °C, shown in Fig. 5c).

Based on these results, we hypothesized that the bubbles of DCPD/AC, -LAC foams were produced by the formation of an AC-induced internal phase within the frontally polymerizing pDCPD phase. Specifically, we intended to identify the effects of AC CM to the subsequently formed internal morphologies of the foams. As shown in Fig. 3a, c, bubbles were arranged in a splayed pattern across the vertical planes of the foams containing 0.5–4 wt% AC and 2, 4 wt% LAC. According to J. A. Pojman[50], this pattern was indicative to a macroscale phase separation of two reactants in a mixture by their individual crosslinking during FP. To assess the effects of these selective crosslinking to the construct of the polymerized millichannel microstructure, the field-emission scanning electron microscopy (FE-SEM) images of the pores were obtained. Unlike the rather rough cross-sectional surface of neat DCPD solid (Fig. 4a), millichannel walls of DCPD/AC foams exhibited smooth curvatures (Fig. 4b–e) that were also observed from the previously reported FP-mediated foams produced by synchronized reactions of DCPD/1,5-cyclooctadiene[22] (ROMP), and -cyclohexane[24] (decomposition). Therefore, we presumed that the nucleated millichannel surfaces were produced by a simultaneous reaction of ROMP and CM of DCPD and AC, respectively. Individually, the spiral-shaped millichannels, which were more clearly observed at 0.5, 1 wt% AC (Figs. 3a, 4b, S11a-b), were representative of the single-head spin mode in FP that emerged from two distinctive crosslinking reactions of different C=C moieties in a mixture. In this mode, an unstable two-dimensional (2D)-front simultaneously moved to its perpendicular directions by the temperature perturbation[50]. According to previous studies, spin mode arose from three contributing factors: the initial reaction temperature, the reactor geometry, and the changes in crosslinking degree during the reaction[50–52]. Since the initial experimental temperature remained constant at RT, we first investigated the effects of changing the reactor diameter to the spin-mode-induced formation of millichannels. FP of DCPD/0.5 wt% AC was separately performed in a glass vial and a borosilicate capillary tube with the diameters ($d$) of 25 mm and 1.5 mm, respectively. The produced millichannels were examined via the FE-SEM and the digital optical microscopy according to the thicknesses of the produced foams. Nevertheless, despite the variation of reactor $d$, the relatively periodic spiral shape (170 ± 41 μm for $d = 25$ mm; 230 ± 32 μm for $d = 1.5$ mm) induced by spin mode was clearly observed from the millichannels (Supplementary Fig. 11a, b). Therefore, these results suggested that the reactor geometry was not a leading contributor to the spin mode in the frontal production DCPD/AC foams.

On the other hand, a directly proportional trend of AC wt% and millichannel width was observed by the merging of adjacent millichannels (Supplementary Fig. 11b–e). This indicated that the changes in the crosslinking degree induced by the CM of AC C=C moieties affected the constructs of DCPD/AC foam morphologies. Consequently, we hypothesized that the reaction dynamics of a conventional FP of DCPD was altered by ACs. To verify this claim, we sought to measure the kinetics of DCPD/AC bulk polymerization at the reaction front[16,53,54]. Therefore, the in situ changes in front velocity ($v_f$) and the maximum

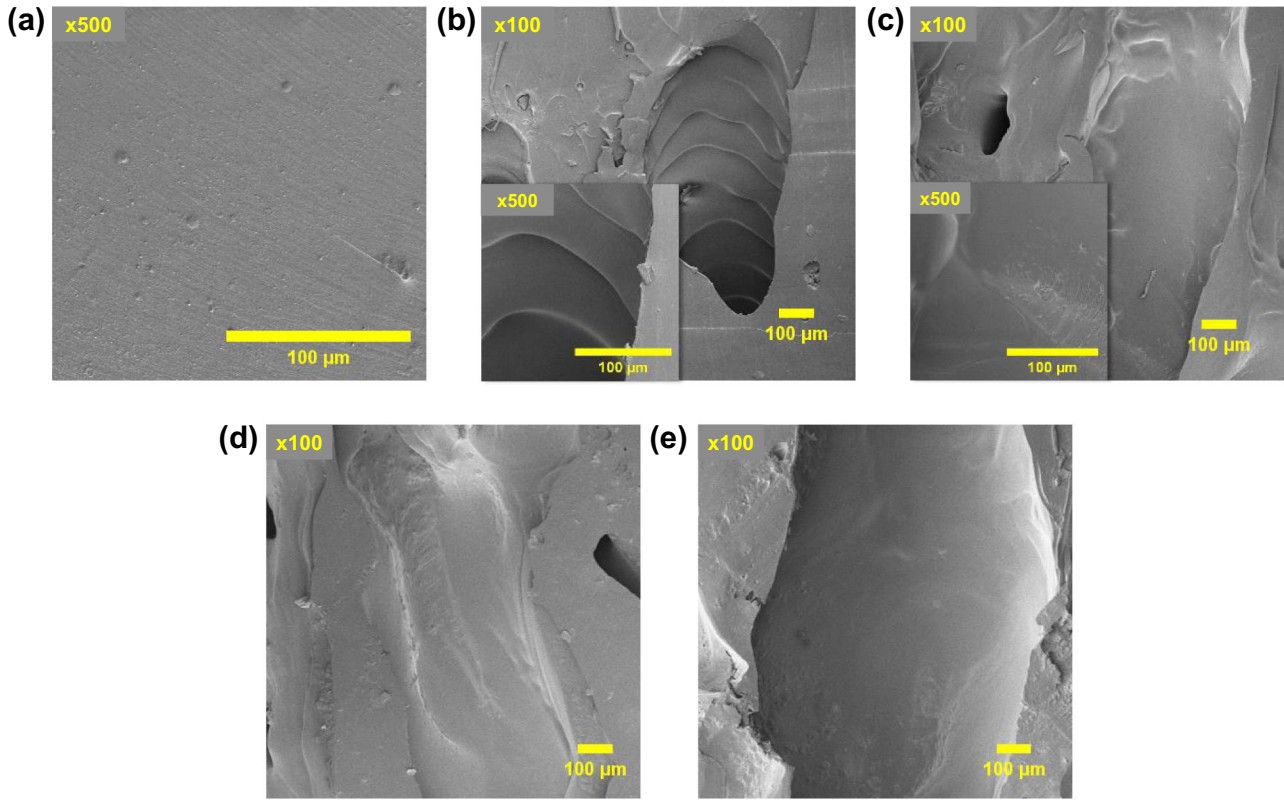

**Fig. 4 Field-emission scanning electron microscope (FE-SEM) images of the produced DCPD/AC foams. a** Neat DCPD solid, (**b**) DCPD/0.5 wt% AC, (**c**) DCPD/1 wt% AC, (**d**) DCPD/2 wt% AC, and (**e**) DCPD/4 wt% AC at ×100–500 magnification.

reaction temperatures ($T_{Max}$) were obtained during the FP of the monomer/AC mixtures. Front displacements (**x**) were first measured by monitoring the movement of front positions across the test tubes that were prelabeled at 1 cm intervals. The obtained **x** values were plotted against time ($t$) to calculate $\mathbf{v}_f$ of each reaction. $T_{Max}$ values were obtained using T-type thermocouples that were submerged in the bottom of the monomer/AC mixtures from which FP was initiated by a localized thermal stimulus and cured the mixtures as propagating fronts traveled along the direction of the transferring thermal energy (Fig. 5a). The **x** vs. $t$ plot of the DCPD/AC specimens showed stable, linearly increasing graphs that confirmed pure FPs (Fig. 5b). As shown in Table 3, $\mathbf{v}_f$ steadily increased from $0.89 \pm 0.2$ mm s$^{-1}$ to $1.7 \pm 0.1$ mm s$^{-1}$ until 2 wt % AC, then slightly decreased to $1.1 \pm 0.1$ mm s$^{-1}$ at 4 wt% AC. Given that % allylation of AC was 79%, this relative decrease in $\mathbf{v}_f$ may have resulted from a mild interruption in the CM reaction due to the increased amount of non-allylated CNC hydroxyl group[55]. Nevertheless, previous systems attributed an increasing trend in $\mathbf{v}_f$ to the increased crosslinking degree that was dependent on the amount of C=C moieties in FP[56,57]. To demonstrate this relationship between C=C moieties and $\mathbf{v}_f$, the reaction velocities of DCPD/AC, -LAC, and -CTD, were compared. As shown in Supplementary Fig. 12a, the introduction of allylated CNC clearly increased the $\mathbf{v}_f$ of neat DCPD, while the addition of CTD significantly decreased $\mathbf{v}_f$ presumably due to the hindered reaction initiation by its non-allylic moieties. Variation of % allylation also affected the reaction dynamics. Unlike from the case of DCPD/AC where $\mathbf{v}_f$ was directly proportional to AC wt%, $\mathbf{v}_f$ obtained from DCPD/LAC specimens exhibited negligible differences from each other. Moreover, even at the lowest wt% of AC, LAC, and CTD, the effect of controlling the amount of C=C moieties to the $\mathbf{v}_f$ of

DCPD FP was clearly demonstrated (Supplementary Fig. 12b). Collectively, the incorporation of highly reactive olefins like the allyl-moieties increased the reaction velocity and was further increased upon their compositional increment. As shown in Fig. 5c, the temperature vs. $t$ profile of neat DCPD and DCPD/ 0.5-4 wt% AC FP reactions were characterized by sharp temperature spikes (i.e., $T_{Max}$) that were observed shortly after the reaction initiations. Without the application of localized stimulus, the reaction temperature remained rather constant at around 21 °C. The flatter regions of the graphs were characterized as the reaction initiation time ($t_{init}$) or the amount of time the initiation zone was heated prior to the advent of a propagating front. A generally increasing trend of $t_{init}$ was observed with increasing AC wt%; however, a dramatic drop in $t_{init}$ at 2 wt% AC was highlighted (Table 3). Nonetheless, according to previous reports, this typical trend of increasing $t_{init}$ was indicative to the lowered reaction reactivity by the introduction of secondary reactants in FP[24,58]. Especially, micro-/nanoparticles behaved like heat sinks that absorbed the heat released from reactions and subsequently decreased $T_{Max}$ by the effects of their shapes and dimensions[21,56,59]. Unlike these reports, the $T_{Max}$ values obtained from the current study exhibited little differences from each other at around 151 °C (Table 3). Therefore, we first investigated the potential relationship between the AC size and $T_{Max}$. However, as confirmed by the TEM analysis (Supplementary Fig. 6), the dimension (width, $w$ x length, $l$) of AC ($14 \pm 3 \times 88 \pm 13$ nm, aspect ratio ~ $6.6 \pm 2$) was in the vicinity of the spherical $SiO_2$ NP width values that were found in the previous FP systems reported by S. P. Davtyan et al.[59] ($w$ ~ 10 nm) and S. Chen et al.[56] ($w$ ~ 20 nm). Meanwhile, a recent study by L. M. Dean et al.[19] reported an enhancement of FP kinetics such as thermal conductivity, $T_{Max}$, $t_{init}$, and $\mathbf{v}_f$ by the increased aspect ratio of

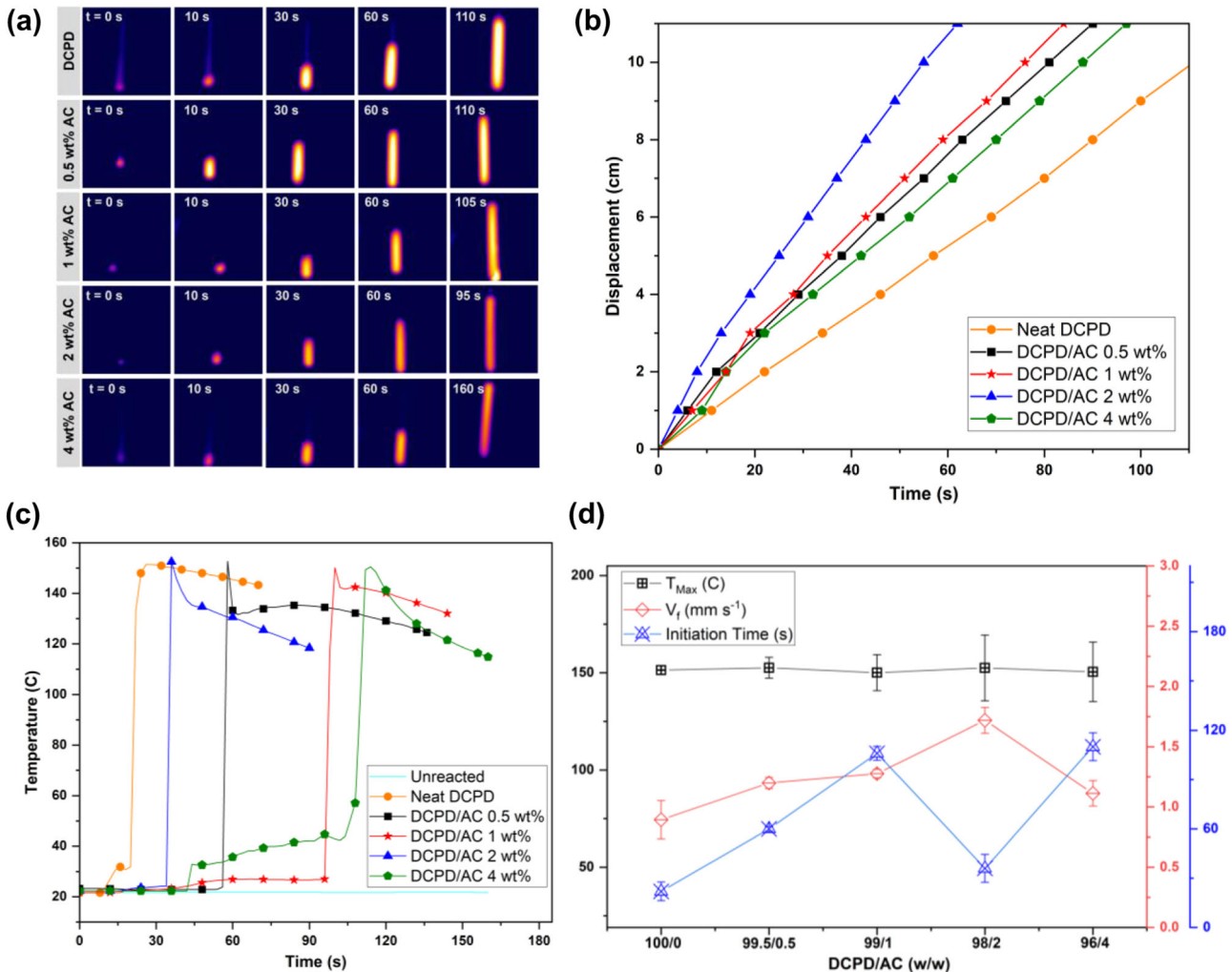

**Fig. 5 Evaluation of the reaction kinetics observed during FP of the DCPD/AC mixtures. a** For increasing AC wt%, the reaction lasted for shorter amounts of time. **b** Front velocities of each reaction were obtained by plotting front displacement against time. **c** The temperature vs. time plots exhibited negligible changes in maximum temperature with increasing AC wt%. On the other hand, a general trend of increasing reaction initiation time was observed with increasing AC wt%, with the exception of 2 wt% AC. **d** Reaction kinetics of 2wt% AC was highlighted as it exhibited the most rapid front velocity and shorter reaction initiation time, respectively, amongst the tested specimens. Standard deviations of the obtained values ($T_{Max}$, $\mathbf{v_f}$, and $t_{init}$) are shown as error bars.

**Table 3 Front velocity ($\mathbf{v_f}$), maximum temperature ($T_{Max}$), and initiation time ($t_{init}$) values were obtained during frontal polymerization of the DCPD/AC samples.**

|  | 0 wt% | 0.5 wt% | 1 wt% | 2 wt% | 4 wt% |
|---|---|---|---|---|---|
| $\mathbf{v_f}$ (mm s$^{-1}$) | 0.89 ± 0.2 | 1.2 ± 0.05 | 1.3 ± 0.03 | 1.7 ± 0.1 | 1.1 ± 0.1 |
| $T_{Max}$ (°C) | 151 ± 0.20 | 153 ± 5.4 | 150 ± 9.3 | 153 ± 17 | 151 ± 15 |
| $t_{init}$ (s) | 22 ± 6 | 60 ± 1 | 106 ± 4 | 36 ± 8 | 110 ± 8 |

The standard deviations were obtained across three replicates of the experiments.

carbon-based NPs. Based on these previous findings, the negligible shifts of $T_{Max}$ in our study may have resulted by the offsetting effect between AC's thermally insulating nature and their relatively high aspect ratio. As for the rather increasing standard deviations of $T_{Max}$ (Table 3), we presumed that an in situ detection of an instantaneous $T_{Max}$ at the pDCPD front was hampered by the stochastic formation of bubbles during DCPD/AC FP, especially by the simultaneously generated air-filled voids (Supplementary Fig. 13a). This was demonstrated by the µ-CT imaging technique that revealed the

locations of the thermocouple hot junctions within DCPD/AC foam pores (Supplementary Fig. 13b–e).

**The influence of ACs to the crystal structure of pDCPD and the resulted relative enhancement of oxidation resistance.** It was revealed from previous studies that the complexity of the DCPD FP cure kinetics was determined by the changes induced to its reaction dynamics (e.g., $T_{Max}$, $v_f$, and $t_{init}$)[54,60]. Especially, these changes that arose from the addition of secondary reactants such as reinforcement fillers[17,54], monomers[22,24,61], and micro-/nanoparticles[19,21] led to the realizations of enhanced mechanical properties, macroscopic patterns, and structural local order in the pDCPD products. In the case of current study, we observed an enhanced oxidation resistance from the produced DCPD/AC foams. Specifically, we hypothesized that by the introduction of 2 wt% AC, an oxidation-resistive crystal structure was polymerized in DCPD foams by its rapid $t_{init}$ (36 ± 8 s) and $v_f$ (1.7 ± 0.1 mm s$^{-1}$) that tandem altered the curing kinetics of FP (Fig. 5d).

Firstly, we observed a gradual discoloration of the DCPD/AC foams from yellowish white to brown over 8 weeks post-production (Supplementary Fig. 14). A typical sign of material

degradation in ambience, as described by S. Kovačič et al.[36,62], oxidation of pDCPD posed a major challenge to the production of porous DCPD monoliths and membranes by drastically altering the material properties and reducing product applicability. Experimentally, oxidation of DCPD/AC foams was confirmed by the normalized ATR-FTIR spectra from which pronounced C=O and O–H absorption peaks, that agreed with previously reported results[62], were observed at around 1700 cm$^{-1}$ and 3390 cm$^{-1}$, respectively (Fig. 6a). Unlike those of other DCPD/AC formulations, however, C=O and O–H peaks pertaining to DCPD/2 wt% AC exhibited significantly lowered intensities that verified the relative oxidation resistance.

According to previous reports, the susceptibility towards oxidative damage resulted from the high density of alkenes along the pDCPD backbone[62]. Therefore, changing the tacticity of polymerizing monomer by controlling the C=C bond stereo-chemistry or by using robust initiators were favorable measures of preventing pDCPD oxidation[36,62]. So far, hydrogenation was successful in enhancing the oxidation resistance of, for example, low-crosslinked pDCPD aerogels[63] and ROMP-mediated nor-bornene polymers[64]. This effect was corroborated by a recent study by Y. Nakama et al.[64] from which a strong correlation between oxidation resistance and the changes induced to the chain conformations or packing modes in crystal lattices was confirmed by the wide-angle x-ray diffraction measurement. Based on these findings, our group sought to investigate the nanoscale effect of ACs to the development of the DCPD/AC foam crystal structures. Therefore, 2D small angle-/wide-angle X-ray scattering (SAXS, WAXS) analyses were performed to capture the AC-induced long-to-short range order changes in the crystal structures of the prepared specimens[64–66]. Both analyses were performed in the transmission mode; the primary X-ray beams were emitted perpendicularly to the vertical axes of the prepared samples after which the scattered beams and the scattering angles ($2\theta$) were collected (Fig. 6b). The obtained 2D scattering images were transformed into one-dimensional scattering patterns by the azimuthal integration method in which the scattered light signals were averaged along the concentric circle around the incident beam by the scattering vector (**q**) or the azimuth angle ($\psi = 360°$)[67,68]. For the SAXS analysis, neat DCPD and DCPD/AC mixtures were frontally polymerized in separate borosilicate capillary tubes ($d = 1.5$ mm) without changing any experimental procedures. Similar to our aforementioned results, more numbers of larger bubbles were formed with increasing AC wt% (Supplementary Fig. 11b–e). For a clearer assessment of crystalline domain structures formed by AC only, scattering light intensities pertaining to neat DCPD solids were background subtracted from those of DCPD/0.5–4 wt% AC foams. The azimuthal-integrated intensity plot revealed structure peaks of 0.5 wt%, 1 wt%, and 4 wt% AC at approximately $\psi_{0.5wt\%AC} = 29°$, 161°, 178°; $\psi_{1wt\%AC} = 29°$, 152°, 173°; and $\psi_{4wt\%AC} = 75°$, 161°, 173° (Fig. 6c). Unlike the cases of these formulations, the structural peaks of 2 wt% AC exhibited a rather periodic pattern across $\psi \sim 29°–132°$, which was hypothesized to represent a relatively homogeneous distribution of crystalline domains along the vertical axis of the DCPD/2 wt% AC foam. This phenomenon was supported by the scattering intensity vs. scattering vector ($\ln(I(\mathbf{q}))$ vs. **q**) plot of 2 wt% AC. While no intensity peaks were observed at other AC wt%, sharp intensity peaks at $q_{Max} = 0.02846$ Å$^{-1}$ and $q_{Min} = 0.03070$ Å$^{-1}$ were observed for 2 wt% AC (Fig. 6d). By the equation $d_L = 2\pi/q_{max}$[68–70], the crystalline domain spacing ($d_L$) of 2 wt% AC was 3.514 nm. Based on these results, it was understood that a long-range order of crystalline domains was uniquely formed by 2 wt% AC. Then, the effects of introducing ACs to the crystal and lattice structures of DCPD were investigated by the WAXS analysis. Prior to this analysis,

neat DCPD solids and the produced DCPD/AC foams were vertically sliced into rectangular shards with approximately 3-4 mm in thickness. As shown in the azimuthal-integrated WAXS intensity profile, the broad shoulder peak obtained from the DCPD/2 wt% AC foam was indicative of the induced orderedness in its crystal lattices by 2 wt% AC (Figs. 6e, f). Meanwhile, the more apparent influence of controlling AC wt% to the changes in the initial DCPD crystal structures was observed from the scattering intensity vs. scattering angle ($2\theta$) plot (Fig. 6g). As shown in Fig. 6h, $2\theta$ generally shifted to the right as AC wt% increased from 0 wt% to 2 wt% ($2\theta_{Neat\ DCPD} = 16.36°$; $2\theta_{2wt\%\ AC} = 16.88°$), after which it slightly shifted to the left at 4 wt% AC ($2\theta_{4wt\%\ AC} = 16.76°$). Using the Bragg's equation ($\lambda = 2d\cos\theta$) and the Scherrer equation ($D = (K\lambda)/(\beta\cos\theta)$), the values of lattice spacing ($d$-spacing) and crystallite size ($D$) that were related to neat DCPD solid and DCPD/AC foams were calculated, where $\lambda$ was the wavelength of the incident beam, $K$ was the shape factor constant ($K = 0.89$), and $\beta$ was the measured full width at half maximum of the $2\theta$ peaks[64,66,68]. By the values of $d$-spacing$_{2wt\%}$ $_{AC} = 0.5248$ nm and $D_{2wt\%\ AC} = 18.19$ nm, it was understood that, compared to the crystal structures formed by other DCPD/AC formulations, those generated by DCPD/2 wt% AC consisted of the smallest crystallites with the tightest packing of crystal lattices (Table 4).

The long-range order of DCPD/2 wt% AC crystalline domains may have arisen from the rapid reaction dynamics. Previous reports had shown the effects of NP surface chemistry to the nucleation rate and thus the foaming processes of polymeric foams[71]. Especially, under a shear force induced unidirectional polymerization, CNCs directed the production of confined anisotropic structures in effect to the simultaneously altering physicochemical dynamics[72,73]. Based on these studies, it may be hypothesized that, at 2 wt% AC, the rapidly moving front physically aligned ACs in the direction of reaction propagation. From the background subtracted SAXS pattern of 2 wt% AC, we observed the formation of a long-range order ($d_L = 3.514$ nm) in the AC crystalline domains. While increasing AC wt% generally decreased both the $d$-spacing and $D$ values of the polymerized crystal structures, those of DCPD/2 wt% AC comprised of the most tightly packed crystallites. Our SAXS and WAXS results revealed that, compared to other DCPD/AC formulations, an optimal amount of 2 wt% AC induced a more homogenous distribution of neatly packed crystallites across the DCPD/2 wt% AC foam. Leveraging the previously reported results that tightly packed lattice structures suppressed oxygen diffusion and material oxidation[61,62,74], it was understood that the structural changes induced by 2 wt% AC were instrumental to the comparative enhancement of oxidation resistance of the DCPD/2 wt% AC foam. Therefore, a one-pot strategy to relatively inhibit oxidation of pDCPD foams by introducing an optimal amount of 2 wt% AC was presented in this study.

## Conclusion

In summary, we have demonstrated an efficient method of producing solid polymeric foams by frontally polymerizing DCPD/AC mixture. Two allyl-functionalized CNC samples were prepared by the varying % allylation: 79 % (AC) and 13 % (LAC). Both AC and LAC were dispersed in a liquid monomer layer at different wt%: 0.5, 1, 2, 4. FPs of DCPD/AC and -LAC mixtures were initiated by a localized thermal stimulus, and cellular solids with varying pore microstructures were produced accordingly to the AC and LAC wt%. The μ-CT analyses clearly demonstrated the interrelationship between % allylation, AC wt%, and the subsequently produced pore sizes and their distributions. Moreover, with increasing AC wt%, % porosity of the foams increased.

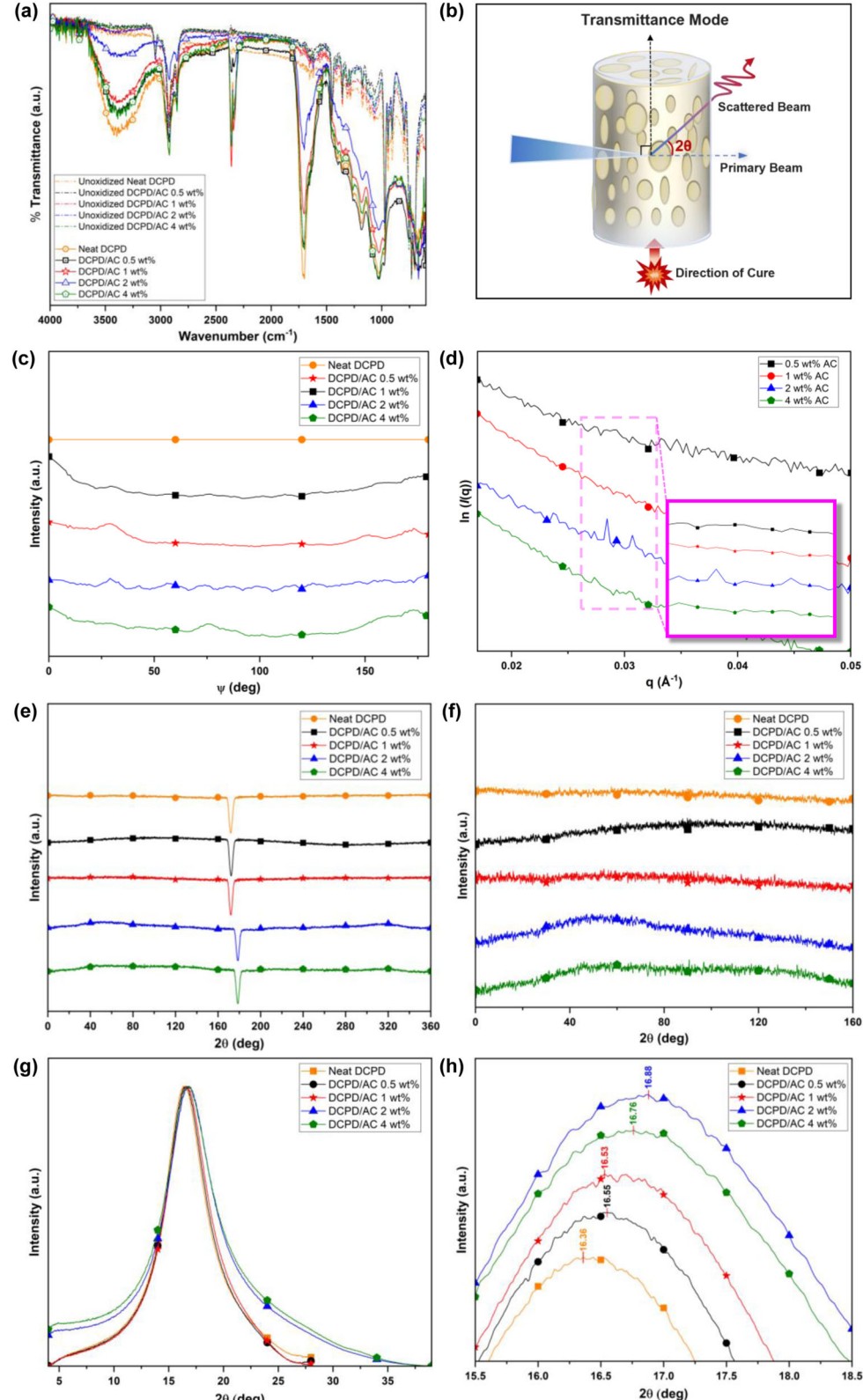

**Fig. 6 An assessment of DCPD/AC foam crystal structures via the ATR-FTIR, SAXS, and WAXS analyses. a** The normalized ATR-FTIR spectra of the prepared neat DCPD and DCPD/AC foams revealing the significantly less oxidized state of DCPD/2 wt% AC foam 8 weeks post-production. **b** A schematic of the transmission mode in the SAXS and WAXS analyses. **c** The background subtracted azimuthal-integrated SAXS pattern of 2wt% AC exhibited a rather homogeneous distribution of crystalline domains between ψ ~ 29°–132°. **d** A long-range order of crystalline domains induced by 2wt% AC was observed by the broad shoulder peak. **e**, **f** WAXS patterns of neat DCPD and DCPD/AC foams suggested that 2 wt% AC induced a directionality in the lattice structures. **g**, **h** The effects of incorporating ACs to the frontally polymerized DCPD crystal structures were observed by the shifts in 2θ.

**Table 4 The shifts in 2θ values, lattice spacing (d-spacing), and crystallite sizes (D) were determined by the WAXS analyses of neat DCPD and DCPD/0.5 – 4wt% AC foams.**

|  | 0 wt% | 0.5 wt% | 1 wt% | 2 wt% | 4 wt% |
|---|---|---|---|---|---|
| **2θ** (deg) | 16.36 | 16.52 | 16.58 | 16.88 | 16.76 |
| Lattice Spacing (d-spacing) (nm) | 0.5414 | 0.5362 | 0.5342 | 0.5248 | 0.5286 |
| Crystallite Size (D) (nm) | 23.48 | 20.90 | 20.47 | 18.21 | 19.38 |

The FE-SEM analyses and the obtained $v_f$, $T_{Max}$, and $t_{init}$ values tandem corroborated the effects of varying amount of AC C=C moieties to the constructs of the DCPD/AC foam micro-structures. Amongst the produced DCPD/AC foams, the relative oxidation resistance of DCPD/2 wt% AC foam was highlighted. The background subtracted SAXS pattern of DCPD/2 wt% AC exhibited a formation of a long-range order within its crystalline domains. The apparent AC-induced changes in the crystal structures of pDCPD were confirmed by the WAXS patterns of the DCPD/AC foams. Specifically, 2 wt% AC directed the formation of a DCPD/AC crystal structure that composed of crystallites with the tightest lattice spacing. In conclusion, we have demonstrated that an optimal amount of 2 wt% AC not only directed the formation of honeycomb-like milichannels but also induced a relatively homogeneous distribution of neatly packed crystalline domains across the DCPD/2 wt% AC foam that subsequently realized an enhancement of oxidation resistance.

## Methods

**Materials**. Dicyclopentadiene (DCPD), 5-ethylidene-2-norbonene (ENB), second-generation Grubbs' catalyst (GC2), phenylcyclohexane (PCH), allyl chloride, urea, sodium hydroxide (NaOH), hydrochloric acid (HCl), and decylamine were purchased from Sigma-Aldrich and used as received. Tannic acid was obtained from Alfa Aesar and used without modification. Tributyl phosphite inhibitor (TBP) was purchased from TCI-Sejin CI. Acetone (99.5%) was purchased from Dae Jung Chemical and used without further purification. Cellulose nanocrystal (CNC) was acquired from Nanografi Nano Technology and used as received. A 26V-65W FX-8801 HAKO soldering iron was used to trigger FP. ISTEK Multimeter CP-500L was used to adjust pH levels. 15 × 150 mm glass test tubes (Tube 2, 15150) were purchased from Dai Han Scientific. 1.5 × 80 mm Borokapillaren capillary tubes were purchased from Muller GmbH. T-type welded thermocouples were purchased from Omega Engineering. TM947SD digital thermometer from Lutron Instruments was used to record temperature values during the FP reaction.

**Preparation of allyl-functionalized cellulose nanocrystal**. AC was synthesized following previous reported procedures[38]. Briefly, 2 g of CNC was dispersed in 50 g of NaOH/urea solvent that was filtered with 5 C filter paper, and consisted of 60 g of NaOH, 40 g of urea, and 1000 mL of dilute water. After being sonicated for approximately 30 min, the CNC-solvent mixture was stored in a freezer (−4 to −10 °C) for 12 h. The frozen solid was thawed and rigorously stirred at RT. For one anhydroglucose unit (AGU) of CNC, 1 to 24 molar equivalence of allyl-chloride was added and was extensively stirred at 35 °C for 72 h under the exclusion of light. The product was neutralized with HCl aq. and precipitated with acetone. The precipitate was further washed with acetone and distilled water three times each, and then freeze-dried at −40 °C overnight to obtain purified AC (Il-Shin Lab Bondiro). The obtained AC was stored at −15 °C under the exclusion of light.

**Formulation of DCPD/AC foam via frontal polymerization**. The DCPD/ENB mixture was made following previously reported systems[15]. DCPD is solid at RT; it was first melted in an oven at 50 °C and blended with 5 wt% of ENB to depress its freezing point. Unless otherwise specified, all references to DCPD refer to the 95/5 DCPD/ENB solution. Five different DCPD/AC mixtures were made by dispersing ACs of varying wt%: 0, 0.5, 1, 2, and 4. The liquid DCPD/AC mixtures were ultrasonicated (Sonics & Materials, VCX 750) for approximately 20 min at amplitude (amp.) of 25 % and 10 min at amp. = 35%, and further sonicated for 10 min (NEXUL NXPC). This mixture was then degassed overnight. Typically, a catalyst/inhibitor solution was made by dissolving GC2 (1E-4 molar equivalence to DCPD) in TBP (2 molar equivalent to GC2)/PCH (50 μL per 1 mg of GC2) mixture. The catalyst/inhibitor solution was added to DCPD/AC mixture and stirred homogenously for 2 min. The mixture was frontally polymerized using a soldering iron that was preheated to approximately 250 °C for 30–40 min.

**Front velocity and temperature measurements**. A glass tube was filled with 11 mL of DCPD/AC mixture. FP was initiated by placing a heated soldering iron in a direct contact with the bottom of the glass tube. Prior to the reaction initiation, the T-type welded thermocouples were connected to the TM947SD digital thermometer, and were submerged to the bottom of the glass tubes from where thermal stimulus was applied and the reaction of FP was initiated. Then the temperature values were recorded for every 2 s. A thermal infrared camera (Keysight U5857A) and Canon EOS 550D were used to monitor front propagation. Front propagation images were obtained from Keysight TrueIR Analysis and Reporting Tool software. The front velocity ($v_f$) was calculated by taking the slope of the best-fit line in the front displacement (x) versus time (t) plot.

**Preparation of CTD**. CTD specimen was made following previously reported procedures[37]. In the first step, 2 g of CNC was uniformly dispersed in 150 mL of distilled water. Then, approximately 0.478 g of HEPES was added to the CNC dispersion, NaOH was added to adjust the mixture's pH level to 8.0, and 0.101 g of tannic acid was added to the form a CNC-TA dispersion. After being stirred at RT for 72 h, 2 mL of CNC-TA mixture was saved in a separate glass vial. Then, 4.0 g of decylamine was added; the resulting CNC-TA-DA (CTD) mixture was rigorously stirred for 3 h at RT before it was thoroughly washed with distilled water and acetone. The CNC-TA intermediate mixture underwent similar washing processes. The reaction products were dried in a vacuum oven at 35 °C overnight and stored at RT.

**Characterization**. Unmodified CNC powder was used without further processing. After freeze-drying AC and LAC, the samples were processed into fine powder using a freeze-miller (SPEX 6875D) for further characterizations. For ¹³C solid NMR analyses, approximately 100 mg of both CNC derivative powders were prepared. ¹³C Inverse-gated solid 500 MHz NMR (Bruker Avance II HD) was performed for AC and LAC in ¹³C MAS 10 kHz spinning mode with contact time of 30 ms for 7.55 h and 11.49 h, respectively. ¹³C CP-MAS solid NMR analyses for AC and LAC were conducted on a solid 500 MHz NMR instrument (Bruker Avance II HD) instrument for 14.10 h and 15.41 h, respectively. ¹³C CP-MAS Solid NMR analyses for ACs and LACs were performed in CPTOSS mode with magic angle spinning rate at 5 kHz and contact time of 2 ms. NMR spectra were analyzed using MestReNova software. For transmission electron microscopy (TEM) analysis of AC, approximately 2 wt% of powdered AC was dispersed in DI water. To evaluate the effect of using probe-ultrasonication device (Sonics & Materials, VCX 750), the AC dispersion was sonicated for 20 min at amplitude of 25 % and 5 min at 35 %. TEM carbon grids were glow discharged using a PELCO easiGlow instrument. 5 μL of AC dispersion was then dropped on a TEM carbon grid. Then, AC-on-carbon grid was negatively stained using 2 μL of uranyl acetate. TEM images of AC dispersion were obtained using the Talos L120C (120 kV) instrument. ATR-FTIR spectra of the unmodified CNC, AC, LAC, DCPD/AC, -LAC, -CTD powders were analyzed using the Thermoscientific Nicolet iS5 instrument and OMNIC software. DCPD/AC foam monoliths were characterized using micro-computed tomography (μCT) (Bruker, Skyscan 1273). They were individually scanned for approximately 25 min at the charge voltage of 60 kV. The transverse, coronal, and sagittal μ-CT images of the scanned samples were obtained using the AccuCT microCT software. % Porosity and degree of anisotropy (MIL = 0.8) across the coronal planes were obtained using the Bruker CTAn software. 550 slices of μ-CT images were analyzed using a rectangular ROI (w x l, 750 pixels x 800 pixels) for both analyses. Average pore diameters were obtained by examining the transverse μ-CT images on the imageJ software. DCPD/AC foams were sliced into smaller pieces with an approximate thickness of 2 mm using stainless blades ahead of the field-emission scanning electron microscope (FE-SEM) analyses (Carl Zeiss, Sigma). The cut specimens were placed on putters and sputter-coated with platinum for 120 s at 60 mA in a vacuum environment using the Leica microsystems instrument. Small angle X-ray spectrometry was performed using the Xenocs XEUSS2.0 instrument. 5 mL liquid DCPD/AC mixtures were put in the 1.5 × 80 mm capillary tubes (Muller GmbH Borokapillaren) and were frontally polymerized by putting a heated soldering iron in contact with the outside of the capillary tubes. For the SAXS analyses, the distance was 2500 mm, and the scattering wavelength and time range were 1.54189 Å and between 600 and 1200 s, respectively. Milichannels that were formed inside the same capillary tubes were observed using the M*i*-9100-ZOOM S digital optical microscopy, and the images were obtained using the UBCAMPRO PROVIX software. Wide-angle X-ray spectrometry was performed using the Bruker D8 Discover instrument across 2θ range of 4–39°. The tested specimens were prepared into shards with approximately 3 mm in thickness. The distance between the detector and the samples was 6 cm, and the scattering wavelength and analysis duration were 1.5418 Å and 600 s, respectively.

## Data availability

The datasets generated during and/or analyzed during the current study are available within the paper and its Supplementary Information file. Supplementary Videos 1, 2, 3, 4 that show the FP reaction of DCPD/0.5-4 wt% AC of this study are available as supplementary video files. The supplementary information contains: ATR-FTIR spectra

of the produced CTD specimen; μ-CT images of the transversal, coronal, and sagittal planes of neat DCPD and DCPD/1 wt% CTD monoliths; Images of DCPD/2, 4 wt% CTD reactions; Peak integrated IG-NMR spectrum of AC; Peak integrated IG-NMR spectrum of LAC; TEM images showing the effect of probe ultrasonicator before the FP of DCPD/AC mixture; Physical characteristics of the formed DCPD/AC foams; Physical characteristics of the formed DCPD/LAC solids; A schematic reasoning the relative decrease in degree of anisotropy shown in this study; Solid-state $^{13}C$ CP-MAS NMR spectra of unmodified CNC, AC, LAC, DCPD, and DCPD/AC foams; XRD patterns of unmodified CNC, AC, LAC, DCPD, and DCPD/AC foams; TGA patterns of unmodified CNC, CTD, AC, and LAC; An FE-SEM image DCPD/0.5 wt% AC that was cured in a 25 mm vial, and digital OM images of DCPD/0.5-4 wt% AC that was cured in 1.5 mm capillary tubes; Front displacement vs. time plot of neat DCPD, DCPD/0.5-4 wt% AC, DCPD/0.5-4 wt% LAC, and DCPD/0.5, 1 wt% CTD reactions; A schematic and μ-CT images showing the positions of thermocouple hot junctions in the produced DCPD/AC foams; Images showing the oxidation of DCPD/AC foams over 8 weeks; Unedited $^{13}C$ CP-MAS Solid NMR spectra and Solid-state $^{13}C$ IG-NMR spectra of AC and LAC.

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

## Acknowledgements

This work was supported by the National Research Foundation of Korea (NRF) grant funded by the Korean government (MSIT) (2020R1A2C1005394) and SNU Materials Education/Research Division for Creative Global Leaders (No. 4120200513611). We greatly appreciate Professor Youngeun Kim, Ph.D from the Materials Science and Engineering Department at Seoul National University for her insightful comments in interpreting our SAXS/WAXS results.

## Author contributions

J.P. conceived and designed the project. S.-Y.K. supervised the project. Syntheses and analyses in the experiments were performed by J.P.

## Competing interests

The authors declare no competing interests.

## Additional information

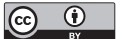

