## [Peer Review File · Communications Chemistry]

Frontal Polymerization-triggered Simultaneous Ring-opening Metathesis Polymerization and Cross Metathesis Affords Anisotropic Macroporous Dicyclopentadiene Cellulose Nanocrystal FoamReviewers' comments:

Reviewer #1 (Remarks to the Author):

This is a nice paper about the use of FROMP to obtain polyDCPD foams by a novel approach. The manuscript is well written, data are well presented and consistent with the discussion and conclusions. The authors investigated many parameters, including composition and reactor geometry. I do recommend it for publication after some minor modifications. Actually, the authors should clarify these points:

l. 147 - the chemical formula would help

l. 148 - FP cannot exhibit a catalytic activity (IUPAC definition)

Table 1 and text - the use of so many decimal digits in non-quantitative data is a non-sense

l. 187 - was the more used descending mode (ignition at the top) tried? Why using the ascending one?

ls. 264,265 - the number of significant digits is not consistent (errors cannot have two decimal digits while the averages have just one)

Fig. 5a - pDCPD solid is not a correct way to distinguish this sample from the others. Actually, all of them are solid.

l. 281 - while V_f and T_{max} are parameters well known in FP, they should need a better definition for readers who are not familiar with the technique.

l. 282 - how reliable long distance IR measurement are? I suspect that they could be affected by the glass surface; in addition, temperature taken inside the sample instead of the external wall should be more significant.

l. 303 - Are all the digits significant?

Fig. 6b - perfectly straight lines are displayed, thus indicating that no deviation from linearity was observed, no experimental error affected the measurement, hundreds of measurements, each one taken a very short time, was performed. This sound strange. Are the displayed lines "real" or are they the interpolations of several points that are not shown?

Reviewer #2 (Remarks to the Author):

In the article, Anisotropic Macroporous Dicyclopentadiene / Cellulose Nanocrystal Foam via Frontal Polymerization-triggered Simultaneous Ring-opening Metathesis Polymerization and Cross Metathesis, Park and Kwak explored the use of cellulose nanocrystals in the synthesis of poly-dicyclopentadiene anisotropic foams using frontal polymerization. The authors claimed that an allyl-functionalized cellulose nanocrystal will participate in a cross-metathesis reaction during the polymerization of dicyclopentadiene. This parallel reaction will improve the phase separation between the two components and as a result the control of pore morphology. The authors synthesized and characterized two allyl-functionalized cellulose nanocrystal batches. Then, they frontally polymerized dicyclopentadiene in the presence of the nanocrystals. Morphological characterization of the foams showed a variation of the pore size as a function of the loading of nanocrystals in the formulation. The authors also studied the influence of the nanocrystals on frontal velocity and maximum temperature of the system. SAXS analyses were used to study the alignment of the cellulose nanocrystals in the foam. Finally, the authors proposed a mechanism for the formation of the foam that involves the chirality of the nanocrystals and their use as nucleating agents.

Major concerns

1. The main claim of the manuscript about the synthesis of anisotropic macroporous dicyclopentadiene/cellulose nanocrystal foams is mainly supported by literature precedent instead of performed experiments.
2. The relation between the foaming process and phase separation caused by the cross-metathesis reaction is not well-explained and supported by experiments.
3. Although the authors claim the formation of anisotropic foams, the alignment of the voids was not

determined.

4. Authors need more control experiments to support the multiple claims in their manuscript.

Minor concerns

1. Line 32-35: The authors claim that understanding the relation between stereochemistry and crystal structures during FP will be useful to enhance the oxidative stability of pDCPD. It is hard to follow the logic of this claim and is beyond the scope of the current publication, which is about the use of cellulose nanocrystals to make pDCPD foams.

2. Line 51-52: Authors should describe what characteristics are they referring to.

3. Line 107-110: The authors claim that an effective internal phase separation would occur caused by the difference in the rate of DCPD crosslinking and nanocrystals cross-metathesis. However, there are no experiments in the paper related to the ROMP and CM rates.

4. Line 139-141: Is 1 wt% enough to induce the formation of porous structures? In the following sections the authors used up to 4 wt% of allylic-modified cellulose nanocrystals, it would be important to show the results with 4 wt% of the CTD-modified cellulose nanocrystals.

5. Line 154: Authors need to explain what an IG-decoupling sequence is.

6. Line 154-166: The authors correctly stated that CP-MAS NMR is not quantitative. However, it is hard to follow their use of this technique to quantify the percentage of allylation in the LAC sample. First, in order to compare the relative intensities of ¹³C-NMR signals across different samples is better to use an internal standard instead of the other cellulose signals. Second, did the authors take the IG NMR spectrum for LAC? What was the result? Third, significant figures of the ratios and percentages need to be double-checked. Fourth, there is a typo in line 166, instead of Table S1 it should say Table 1.

7. Line 178: The claim needs a reference.

8. Line 181: It is unclear why the authors studied the agglomeration in aqueous media instead of the monomer.

9. Line 193-197: If the authors started with the same mass in all the experiments, why did they observe a decrease in the mass after polymerization? It is not clear why by using a higher percentage of AC the authors obtained foams with lower mass.

10. Line 200: The authors claim the directional anisotropy is maintained. However, there are no calculations that support this claim.

11. Line 202-208: The authors suggest LAC performs worse than AC in forming polymeric solid foams. However, this claim is based solely on the comparison of images 4a and 4b. Authors need to calculate alignment, average diameter, and distribution to support this claim.

12. Line 224: It is not clear the use of the arrow to indicate the direction of propagation in Figures 4a and 4b. The figures suggest the authors are showing not just the parallel section but also the transversal one.

13. Line 228: Double-check the significant figures in Table 2 and reconsider the importance of the inclusion of R2.

14. Line 231-232: Reconsider rephrasing for more clarity.

15. Line 231-233: The claim about the no crosslinking between DCPD and AC is hard to support by ¹³C-NMR because at these concentrations the AC signals are not observed in Figure S5. Therefore, it is highly unlikely to observe a signal corresponding to cross-metathesis in the pDCPD signal around 130 ppm.

16. Line 237-239: The authors suggest that bubbles are forming from a phase separation phenomenon. It would be important that the authors expand this concept. Where are the bubbles coming from? Are they formed from dissolved gases? Why phase separation produces bubbles in this system?

17. Line 239-250: The authors claim that the wrinkles observed in the foams are caused by the polymerization rate difference between ROMP and CM. However, as the authors stated, this behavior had been observed in another system where there is only one type of reaction (ROMP). Therefore, the claim can't be supported by the experiment.

18. Line 248-267: The authors discussed the observation of spin mode in formulations with low content of AC. However, no connection of the spin mode with the hypothesis of the manuscript was

provided. It is hard to understand the reason why this observation was included in the manuscript. Moreover, Figures 4a, 4b, and S3a-b are not related to the study of the spin mode. Why was spin mode not observed in the other formulations? Why were two different imaging techniques used to compare Figure S6?

19. Line 274: The authors should include the images of other formulations together with Figure S6c to support the increase in the multichannel size.

20. Line 282: The use of an IR camera to measure maximum temperature is not accurate because glass absorbs IR radiation.

21. Line 290-298: Authors suggest the increase in frontal velocity is due to an increase in the amount of monomers containing C=C moieties. To support this claim authors need to demonstrate that the amount of C=C coming from the AC is higher than the amount of replaced C=C from DCPD. Additionally, this suggestion does not explain why the 4 wt% has a lower frontal velocity. Claim about allyl-moieties being more reactive than cyclopentene is misleading because dicyclopentadiene contains both high strain and low strain cyclic alkenes. Therefore, a discussion about the low reactivity of one particular moiety cannot be done separately.

22. Line 299-301: Discussion about the relation between monomer consumption and frontal velocity is hard to follow and does not explain the other results.

23. Line 313: Values in the table need standard deviation

24. Line 336: What are the intensities the authors are integrating?

25. Line 337-338: The claim about the increase in anisotropy by means of crystallinity cannot be supported by Figure 7a. There are no calculations included about the crystallinity and how it changes by increasing the amount of AC. Additionally, LAC samples do not show any peaks that allow the determination of anisotropy.

26. Line 339: Figures S3a and S3c do not correspond to the current discussion.

27. Line 339-347: Authors suggest that polydispersity is problematic in the determination of crystallinity via SAXS. However, they claim the samples are more crystalline which indicates a higher anisotropy. If SAXS generated unclear results about the alignment of the AC, the authors need to include other experiments that corroborate their assumption.

28. Line 346-349: Authors need to expand on the relation between DCPD oxidation and the foaming mechanism.

29. Line 353: It is hard to see the difference in oxidation with the images provided in Figure 7b. FTIR characterization will be helpful to corroborate their differences.

30. Line 369-372: How do the authors know that an organization of the polymerizing DCPD is happening in their system? It is hard to support this claim just by referencing literature on other systems, which are dramatically different from the current one.

31. Line 372-375: Authors need to provide quantitative oxidation and crystallinity values for all the samples to support that they are correlated with each other.

32. Line 383: How do the authors know that the 2wt% mixture has a stoichiometric amount of functional C=C and chiral-inducing hydroxyl groups?

33. Line 385-387: The authors use the word "synergistically" to explain their claims about the organization of the CN. However, a better discussion is needed to support this hypothesis.

34. Line 400-418: Conclusions need to be rewritten based on the suggestion made along with the manuscript revision.

Decision: Too preliminary

This is interesting work but the authors need to address major issues in the manuscript to better support the multiple claims they made. Additionally, seems that the authors tried to explain some of their observations with highly complex hypotheses, mostly derived from the work of other authors. Instead, authors should propose simpler explanations that fit better their system.

Reviewer #3 (Remarks to the Author):

The manuscript by Jinsu Park et al reports a rapid production of solid polymeric foams by frontally polymerizing DCPD/AC mixture. DCPD/AC foams with varying pore microstructures were produced. The catalyst chemistry governing FP triggered a simultaneous reaction of ring-opening metathesis polymerization (ROMP) and cross metathesis (CM) of DCPD and AC, respectively, and induced a robust internal phase separation. Although this work is interesting and well writing. However, there are some issues should be addressed. This manuscript, at least the present form is not suitable for publication.

1. The authors state that "For increasing AC wt%, paler and more porous foams were produced (Figure 3a), but it is not obvious from the figure. High quality image should be provided.
2. In FP process, the temperature profile is highlighted with a stable horizontal line and a maximum point. This temperature profile should be tested at an unreacted fixed point. Thus, the Figure 6c might be retested.
3. The authors state that "the more obvious sign of DCPD oxidation in air is the color change from white to brown. This is not rigorous. Some quantitative analysis should be conducted.
4. The proposed mechanism needs to be further elucidated.

[Response to REVIEWER #1 Comments]

Dear Reviewer #1,

We are incredibly grateful your thoughtful advice and comments to our manuscript. Our group firmly believe that your insightful suggestions have led us to significantly improve the depth of our work. Once again, we would love to express our gratitude for your time and consideration.

Thank you so much.

Sincerely,

Prof. Seung-Yeop Kwak

l. 147 - the chemical formula would help

The reviewer advised to provide the chemical formulae to better help readers understand the synthesized cellulose structure. We most definitely agree with the reviewer's advice and edited it accordingly.

Relevant Section in the New Manuscript: Lines 145 - 147

Leveraging this catalyst chemistry, we postulated that a more effective separation of monomer-AC phases may be achieved by introducing additional C=C moieties ($R'CH_2CH=CH_2$) to the CNC ($C_6H_7O_5$) surface chemistry, *i.e.*, by using AC ($C_6H_7O_5R_3$, $R = H$ or $CH_2CH=CH_2$)³⁸.

l. 148 - FP cannot exhibit a catalytic activity (IUPAC definition)

The reviewer advised to provide the chemical formulae to better help readers understand the synthesized cellulose structure. We most definitely agree with the reviewer's advice and edited it accordingly.

Relevant Section in the New Manuscript: Line 147 - 149

Specifically, we hypothesized that the C=C moieties on AC would also react to GC2 and trigger a simultaneous ROMP and CM of DCPD and AC, respectively.

Table 1 and text - the use of so many decimal digits in non-quantitative data is a non-sense

The reviewer has kindly pointed out the significance of the non-quantitative data obtained from solid state ¹³C CP-MAS NMR analyses of allyl-functionalized cellulose

nanocrystal (AC) and low allyl-AC (LAC). Previously, these values were used to approximate the % allylation of AC and LAC. However, we entirely agree that a more precise quantitative data should be provided to calculate % allylation. Therefore, we have conducted ^{13}C Inverse-gated solid NMR (IG-NMR) for LAC sample and obtained its quantitative peak integration values.

In the current IG-NMR analyses, carbon peak at C1 position in the AC AGU unit was integrated as the standard peak (1.00). The integrated carbons peak values that were assigned to allyl-groups in AC were 0.1363 (C7), 0.9210 (C8), and 1.3199 (C9). Those in LAC were 0.1325 (C7), 0.2159 (C8), and 0.0441 (C9). Using the typical method of calculating degree of substitution of cellulose^{42,43}, a complete allylation or the sum of peak integration across C7 – 9 positions was assumed to be 3. Therefore, % allylation was obtained by

$$\% \text{ allylation} = \frac{(\text{sum of the obtained peak integrated values of C7, C8, C9})}{3}$$

In this regard, % allylation of AC and LAC are approximately 79.24% and 13.08% respectively.

Unmodified NMR spectra of AC and LAC are provided in Fig. S1 and S2.

Relevant Section in the New Manuscript: Table 1

Table 1. Quantitative allyl/AGU ratios and the approximated % allylation in AC and LAC were obtained via ^{13}C Inverse-gated Solid NMR (IG-NMR).

	Quantitative Allyl/AGU Ratio	% Allylation
Allyl-Cellulose (AC)	2.4:1	79
Low Allyl-AC (LAC)	0.39:1	13

Relevant Section in the New Manuscript: Figures S4, S5

Figure S4. Solid state ¹³C Inverse -gated NMR spectrum of AC. Peak integration values of the carbon peaks pertaining to the allylic group assigned to C7, C8, and C9 were 0.14, 0.92, and 1.3, respectively.

Figure S5. Solid state ¹³C Inverse -gated NMR spectrum spectrum of LAC. Peak integration values of the carbon peaks pertaining to the allylic group assigned to C7, C8, and C9 were 0.13, 0.22, and 0.044, respectively.

I. 187 - was the more used descending mode (ignition at the top) tried? Why using the ascending one?

We appreciate the reviewer's concern over the direction of ignition. We have previously tried initiation of frontal polymerization (FP) from the top of the cylinder. However, there were a couple of aspects relating to the reaction that led us to make the decision of initiating the reaction from the bottom. First, when igniting from the top, our AC-DCPD liquid mixture had to be in direct contact with the soldering iron. In this case, foams started to form from the tip of the soldering iron, and we had to hold the iron until the foams were large enough for the iron to be removed from the unreacted, liquid mixture. We believed that this method did not qualify for a 'localized initiation' in which the thermal stimulus would be removed upon observation of FP initiation. Additionally, our earlier experiments measuring frontal velocity raised the possibility that the reactions initiated from the top and bottom may be different. We provide a front displacement vs time plot that compares different initiation methods in Fig. R1. Although the reaction is unclear, our group concluded that for this study, FP reaction should be initiated from the bottom of the glass tube without making a direct contact between the soldering iron and the AC- or LAC-DCPD liquid mixture.

Figure R1. Front Displacement (cm) vs Time (s) Plot comparing the FP reaction of Neat DCPD initiated from the Top and from the Bottom of the Glass Tube.

ls. 264,265 - the number of significant digits is not consistent (errors cannot have two decimal digits while the averages have just one)

As the reviewer kindly advised, we have revised the relevant section in our newly edited manuscript. We greatly appreciate the reviewer's attention to the details in our manuscript.

Relevant Section in the New Manuscript: p. 14 line 264 - 265

Nevertheless, both results clearly displayed millichannels formed *via* single head spin mode during DCPD/AC FP with relatively consistent periodicity ($230 \pm 32 \mu\text{m}$ for $d = 1.5 \text{ mm}$; $170 \pm 41 \mu\text{m}$ for $d = 25 \text{ mm}$).

Fig. 5a - pDCPD solid is not a correct way to distinguish this sample from the others. Actually, all of them are solid.

We definitely agree with the reviewer's comment that the description for Fig. 5a needed a clearer mode of distinguishing each DCPD solids or foams. Grateful for this advice, we have made an appropriate revision on our new manuscript.

Relevant Section in the New Manuscript: p. 14 Text in Figure 5a.

Figure 2. Field emission scanning electron microscope (FE-SEM) images obtained for (a) neat DCPD solid, (b) DCPD/0.5wt% AC, (c) DCPD/1wt% AC, (d) DCPD/2wt% AC, and (e) DCPD/4wt% AC at x100 – 500 magnification.

1. 281 - while V_f and T_{max} are parameters well known in FP, they should need a better definition for readers who are not familiar with the technique.

We are grateful that the reviewer has kindly raised the necessity to include the definitions of frontal velocity (v_f) and maximum temperature (T_{max}) and their importance in the reaction of FP. Therefore, we have included further elaboration of the two terms in our new manuscript.

Relevant Section in the New Manuscript: Lines 280-282

To investigate the effects of increasing C=C moiety to the dynamics of a GC2-mediated FP reaction, we sought to measure the kinetics at the reaction front of the bulk polymerization of DCPD/AC mixtures^{16,53,54}. Therefore, we observed the changes in front velocity (v_f) and maximum reaction temperatures (T_{Max}) of the prepared monomer-AC mixtures during FP.

1. 282 - how reliable long distance IR measurement are? I suspect that they could be affected by the glass surface; in addition, temperature taken inside the sample instead of the external wall should be more significant.

We most definitely agree with the reviewer that using an IR camera from a distance for T_{max} measurement was not the most accurate method. Thus, we obtained temperature values for every 2 seconds intervals from the inside of the sample using commercially available T-type thermocouples from Omega Engineering. With these new set of data, we provide a more reliable set of T_{Max} values of neat DCPD (AC 0 wt%), and DCPD/AC 0.5, 1, 2, 4 wt%. Briefly, we observed little differences between the T_{Max} values at around 151 °C. This observation was in contrast to previously reported particle-mediated FP systems where NPs, as thermal sinks, partially absorbed the heat released during the reaction.^{1,2,3} The size (width) of NPs seemed to

induce negligible effect to T_{Max} as the size AC ($w \times l$), from our TEM result, was approximately 14 x 88 nm which fell in between the sizes of spherical SiO₂ NPs of 20 nm and 10 nm reported by S. Chen *et al*¹ and S. P. Davtyan *et al*², respectively. Meanwhile, a recent work exhibiting increased T_{Max} for increasing carbon nanotube (CNT) wt% in a DCPD FP system was reported by L. M. Dean *et al*⁴. According to their work, this phenomenon was attributed to the enhanced thermal conductivity during FP induced by the high aspect ratio of CNTs, as well as carbon-based NPs' robust intrinsic photothermal conversion efficiencies.⁴ Based on these previous studies, it may be understood that AC's nonconductivity and high aspect ratio offset each other thus resulted in the small T_{Max} shifts compared to that of neat DCPD.

These findings and the related explanations have been edited into relevant sections in the new manuscript.

References

1. Chen, S., Sui, J., Chen, L. & Pojman J. A. Polyurethane-Nanosilica Hybrid Nanocomposites Synthesized by Frontal Polymerization. *J. Polym. Sci. A Polym. Chem.*, **43**, 1670-1680 (2005).
2. Davtyan, S. P., Berlin, A. A., Shik, K., Tonoyan, A. O. & Rogovina, S. Z. Polymer Nanocomposites with a Uniform Distribution of Nanoparticles in a Polymer Matrix Synthesized by Frontal Polymerization Technique. *Nanotechnol Russia*. **4**, 489-498 (2009).
3. Gao, Y. *et al*. Controllable Frontal polymerization and spontaneous patterning enabled by phase-changing particles. *Small* **17**, 2102217 (2021).
4. Dean, L. M., Ravindra, A., Guo, A. X., Yourdkhani, M. & Sottos, N. R. Photothermal initiation of frontal polymerization using carbon nanoparticles. *ACS Appl. Polym. Mater.* **2**, 4690-4696 (2020).

Relevant Section in the New Manuscript: **Figure 5c, Table 3**

Figure 5c. Maximum Temperature (T_{Max}) (C) vs Time (s) Plot of an Unreacted Reaction of DCPD/AC Mixture, and Thermally Initiated FP Reaction of Neat DCPD, and DCPD/AC 0.5-4wt%.

Table 3. Front velocity (v_f), maximum temperature (T_{Max}), and initiation time ($t_{initiatoin}$) values were obtained during frontal polymerization of the DCPD/AC samples.

	0 wt%	0.5 wt%	1 wt%	2 wt%	4 wt%
V_f (mm s ⁻¹)	0.89 ± 0.2	1.2 ± 0.05	1.3 ± 0.03	1.7 ± 0.1	1.1 ± 0.1
T_{Max} (°C)	151 ± 0.20	153 ± 5.4	150 ± 9.3	153 ± 17	151 ± 15
$t_{initiation}$ (s)	22 ± 6	60 ± 1	106 ± 4	36 ± 8	110 ± 8

Relevant Section in the New Manuscript: Line 291

Fig. 6c shows a new temperature (°C) vs time (s) plot in which temperature peaks i.e., maximum temperature (T_{Max}), were observed from the prepared specimens. The flatter region from each plot represents contact time of the soldering iron to the glass tube before foaming

processes were observed. Without thermal initiation, the temperature of the DCPD/AC mixture remained constant at approximately 22 °C. Upon the application of thermal stimulus, neat DCPD and DCPD/AC 0.5-4wt% samples displayed minimal differences in their T_{Max} values at around 151 °C which falls within the previously reported region of T_{Max} values observed during FP of DCPD.²⁴ On the other hand, earlier works reported that for increasing the wt% of micro-/nanoparticles, T_{Max} decreased during FP of DCPD due to the ability of the filler to absorb heat released from the reaction.^{21, 47}

Relevant Section in the New Manuscript: Line 458-460

T-type welded thermocouples were purchased from Omega Engineering and were used as received. Thermocouples were submerged to the bottom of the glass tubes from where thermal stimulus was applied and the reaction of FP was initiated.

1. 303 - Are all the digits significant?

The reviewer has kindly pointed out to the details of reporting the obtained maximum temperature (T_{Max}) values. With the newly measure T_{Max} values, we addressed our observation in line 291 which was shown in our previous edit above.

Relevant Section in the New Manuscript: p. 15 Line 291

Fig. 6c shows a new temperature (°C) vs time (s) plot in which temperature peaks i.e., maximum temperature (T_{Max}), were observed from the prepared specimens. The flatter region from each plot represents contact time of the soldering iron to the glass tube before foaming processes were observed. Without thermal initiation, the temperature of the DCPD/AC mixture remained constant at approximately 22 °C. Upon the application of thermal stimulus, neat

DCPD and DCPD/AC 0.5-4wt% samples displayed minimal differences in their T_{Max} values at around 151 °C which falls within the previously reported region of T_{Max} values observed during FP of DCPD.²⁴ On the other hand, earlier works reported that for increasing the wt% of micro-/nanoparticles, T_{Max} decreased during FP of DCPD due to the ability of the filler to absorb heat released from the reaction.^{21, 47}

Fig. 6b - perfectly straight lines are displayed, thus indicating that no deviation from linearity was observed, no experimental error affected the measurement, hundreds of measurements, each one taken a very short time, was performed. This sound strange. Are the displayed lines "real" or are they the interpolations of several points that are not shown?

We greatly appreciate the reviewer's concern over Figure 6b we have previously reported. We do agree with the reviewer that the interpolated plots of front displacement (cm) vs time (s) would not clearly convey the credibility of our results while this was not uniformly applied to other subfigures in Figure 6. Therefore, we have replaced the previously reported real values of newly measured front displacement (cm) vs time (s) plot. The sharp changes in the slopes of all plots were the results of less accurate front displacement measurements by the eye due to the curvature in the bottom of glass tubes.

Relevant Section in the New Manuscript: Figure 5b

[Response to REVIEWER #2 Comments]

Dear Reviewer #2,

We are greatly honored to have received your comments and suggestions. Our group are ascertained that the overall quality of this work has significantly improved while answering the questions the reviewer had raised during this process.

Once again, we are incredibly grateful for your time and consideration, and we would love to express our immense gratitude for the important questions the reviewer had asked that eventually led us to have discovered more in this study.

Thank you so much.

Sincerely,

Prof. Seung-Yeop Kwak

Line 32-35: The authors claim that understanding the relation between stereochemistry and crystal structures during FP will be useful to enhance the oxidative stability of pDCPD. It is hard to follow the logic of this claim and is beyond the scope of the current publication, which is about the use of cellulose nanocrystals to make pDCPD foams.

We greatly appreciate the reviewer's concern over the scope of the current manuscript. With the help of the reviewer's followed comments across our manuscript, the newly found relationship between the formed DCPD/AC foam crystal structures and the unique oxidation resistance of DCPD/2wt% AC foam was highlighted in the later section of our newly edited manuscript. Briefly, by reviewer's suggestion, we have first quantified the levels of oxidized states of the DCPD/AC foams through the ATR-FTIR analysis. Then from the SAXS and WAXS analyses, it was revealed that 2wt% AC induced a formation of a long-range order of its crystalline domains independently from those of DCPD, and that, within the DCPD/2wt% AC foam, a homogeneous distribution of tightly packed crystallite structures relatively suppressed the oxygen diffusion. Based on these findings, the abstract portion of our new manuscript was edited.

Relevant Section in the New Manuscript: Abstract

Multifunctionality and effectiveness of macroporous solid foams in extreme environments have captivated the attention of both academia and myriad industries. The most recent rapid, energy-efficient strategy in manufacturing solid foam with directionality is the frontal polymerization (FP) of dicyclopentadiene (DCPD). However, there still remains the need for a time efficient one-pot strategy to induce anisotropic macroporosity in DCPD foams. Here we show a rapid production of cellular solids by frontally polymerizing a mixture of DCPD monomer and allyl-functionalized cellulose nanocrystals (AC). Our results demonstrated the correlation between increasing % allylation and AC wt%, and the formed

pore architectures. Especially, with the optimal amount of 2wt% AC, more rapid front velocity and reaction initiation time were observed. From the small- and wide-angle X-ray diffraction analyses, it was revealed that the introduction of 2wt% AC affected the crystal structure of frontally polymerized DCPD/AC foams and thus induced a relative resistance to oxidation.

Line 51-52: Authors should describe what characteristics are they referring to.

We certainly agree with the reviewer that the properties rising from the materials microstructures needed much clarity to efficiently convey our contextual messages to the readers.

Relevant Section in the New Manuscript: p. 5 Line 49 – 53

Inspired by nature's structural complexity and topologies, more recent works have investigated different foaming processes such as 3D printing and freeze casting to better control foam architecture and to realize the followed unique characteristics such as improved thermal and fire retardancy, and strength-to-weight ratio, respectively.^{11,12}

References

- (11) Yu, Z.-L. *et al.* Bioinspired polymeric woods. *Sci. Adv.* **4**, eaat7223 (2018).
- (12) Konka, J., Buxadera-Palomero, J., Espanol, M. & Ginebra, M.-P. 3D Printing of hierarchical porous biomimetic hydroxyapatite scaffolds: adding concavities to the convex filaments. *Acta Biomaterialia* **134**, 774-759 (2021).

Line 107-110: The authors claim that an effective internal phase separation would occur caused by the difference in the rate of DCPD crosslinking and nanocrystals cross-metathesis. However, there are no experiments in the paper related to the ROMP and CM rates.

We greatly appreciate the reviewer's thoughtful and critical comment addressing the lack of proof for our claim in our original manuscript.

Based on the reviewer's comment, we obtained the front velocity (v_f) values of neat DCPD, DCPD/AC, DCPD/LAC, and DCPD/CTD to compare the effects of varying % allylation and wt% of the allylated CNCs. Especially, v_f obtained for neat DCPD would be related to the reaction rate of ROMP, those obtained from the DCPD/AC and -LAC reactions would provide us a relationship between the varying % allylation, allylated CNC wt%, and the crosslinking rate by CM. Additionally, v_f obtained from DCPD/CTD reactions, as they represent the reaction rate of a monomer/non-reactive nanoparticle FP system, would clearly demonstrate the role of the allyl moieties of AC in the production of DCPD/AC foams. As shown in Fig. S12a, the drastic differences in v_f between the tested samples were observed. Firstly, compared to v_f of neat DCPD, those of allylated CNCs (i.e., AC and LAC) globally exhibited an increase. On the other hand, v_f that were obtained from DCPD/0.5 – 1wt% CTD reactions decreased with increasing CTD wt%. It should be noted that the reaction with 2, 4wt% CTD was not initiated, and we presume that the larger amount of non-allylic moiety of CTDs rather hindered the initiation of the FP reaction itself (Figure S3). Nevertheless, as shown in Fig. S12b, even with the smallest addition of each CNC variants (AC, LAC, CTD), the differences in v_f were clearly observed. Based on these results, we demonstrated the rate differences of not only ROMP of DCPD but also those of CM by comparing the reactions of DCPD/AC, -LAC, and -CTD.

These findings were applied to the appropriate sections of our newly edited manuscript that are shown below.

Relevant Section in the New Manuscript: **Figure S12**

Figure S12. Front displacement (x) vs. time (t) plots of (a) neat DCPD, DCPD/0.5 – 4wt% AC, DCPD/0.5 – 4wt% LAC, and DCPD/0.5, 1wt% CTD reactions were obtained. When comparing the smallest wt% of each cellulose variants, (b) a significant difference in the measured reaction kinetic was observed.

Relevant Section in the New Manuscript: **p. 18 Lines 334 – 356**

The x vs. t plot showed stable, linearly increasing graphs of the prepared DCPD/AC specimens, indicating pure FPs (Figure 5b). As shown in Table 3, v_f steadily increased from $0.89 \pm 0.2 \text{ mm s}^{-1}$ to $1.7 \pm 0.1 \text{ mm s}^{-1}$ as AC wt% increased up to 2wt%, then slightly decreased to $1.1 \pm 0.1 \text{ mm s}^{-1}$ when it reached 4wt%. The relative decrease in v_f in the case of 4wt% AC may have resulted from the mild interruption in the chain reaction during CM due to the increased amount of non-allylated hydroxyl groups on CNC surfaces, since % allylation in AC did not yield 100% from our result.⁵⁵ Nevertheless, similar trend of increasing v_f with increasing C=C moieties was reported in a number of previously studied FP systems whose feature was attributed to the increased crosslinking degree.^{56,57} Experimentally, this phenomenon may be better understood by comparing the reaction velocities of DCPD/AC, -LAC, and -CTD. Figure S12a depicts the drastic differences in v_f in relations to the presence of allyl groups, % allylation of CNCs, and their wt% in FP reactions. The effect of allylating CNCs in FP can be clearly observed, as compared to neat DCPD, FP of DCPD/AC and -LAC

displayed increased v_f while DCPD/CTD samples exhibited significant decrease in v_f with increasing CTD wt%, presumably due to the hindrance of an uninterrupted reaction by the non-allylic moieties. The impact of % allylation pertaining to the reaction dynamics can also be realized. While FP containing ACs generally showed increasing trend of v_f , those incorporating LACs revealed negligible differences between the measured v_f values. This phenomenon appears clearer when comparing the smallest wt% of AC, LAC, and CTD in FP of DCPD (Figure S12b). In this sense, the effect of incorporating highly reactive olefin metathesis group like allyl-moiety into FP of DCPD was translated into an overall increase in v_f that exhibited further rise upon the compositional increment of reactive C=C bonds.

Line 139-141: Is 1 wt% enough to induce the formation of porous structures? In the following sections the authors used up to 4 wt% of allylic-modified cellulose nanocrystals, it would be important to show the results with 4 wt% of the CTD-modified cellulose nanocrystals.

We appreciate the reviewer's concern over the feasibility of tannic acid-decylamine (CTD) modified CNCs (CTD-CNCs) to inducing porous structures during FP of DCPD. Perhaps our group lacked clarity in describing this portion of our manuscript as we tried to deliver the fact that CTD-CNCs, even at 1 wt%, were not able to induce successful phase separation and produce porous structures. However, based on the reviewer's kind advice, our group performed FP of DCPD/CTD-CNC 4 wt% on two separate occasions. Unfortunately, we were not able to initiate FP reaction of the DCPD monomer-4wt% CTD-CNC mixtures. The first problem our group witnessed was in the dispersion of CTD-CNCs. As seen in Fig. R2a, even after treating the monomer-CTD-CNC mixture at the same amplitudes and over the same duration as we did to the monomer-AC/LAC mixture in the current study, our group observed a difficulty in dispersing CTD-CNC NPs. When initiated from the bottom of the glass tube, the reaction did not occur. On the other hand, when initiated from above, in which the phase mostly

consisted of DCPD monomer, FP was initiated but did not propagate when the reaction front reached the DCPD-CTD layer that was accumulated to the bottom of the glass tube.

To be assured that the size of the added NPs was not the leading factor of this phenomenon, we freeze-milled the CTD-CNC powder produce even-finer powders. With these, our group seemed to have overcome the aforementioned challenge regarding the dispersion of CTD-CNC powders. However, FP was still not initiated when the soldering iron was indirectly in contact with the bottom of the tube. When we dipped the tip of the soldering iron into the liquid monomer-CTD mixture, FP reaction was initiated from the surface of the tip; however, the reaction did not propagate leaving only a partial amount of the monomer-CTD mixture polymerized (Fig. R2c) From these results, our group has reconfirmed that without C=C moiety on CNCs, the production of DCPD foams leveraging its chemistry was not possible. Furthermore, the more the amount of CTD-CNCs was present in a DCPD FP system, the more difficult the reaction initiation. We postulate that the non-allylic moieties decorated on CNC hindered a reaction of uninterrupted frontal ring-opening metathesis polymerization (FROMP) of DCPD monomer.

Relevant Section in the New Manuscript: Figure S3

Figure S3. FP of DCPD/CTD were performed at (a) 2wt% CTD and (b – d) 4wt% CTD. The same thermal stimulus used in the FP reactions of neat DCPD, DCPD/AC, and DCPD/0.5 – 1wt% CTD could not initiate the reaction of 2 – 4wt% CTD through heating of the bottom of the glass. When put the soldering iron in direct contact with the liquid monomer/CTD mixture, gel-like solid was formed; however, the reaction did not propagate thus leaving the unreacted portion in liquid form. (e) A schematic of the generated CTD material may suggest that that higher wt% of the non-allylic moiety of CTD hindered an uninterrupted FP reaction to be initiated.

Line 154: Authors need to explain what an IG-decoupling sequence is.

We agree with the reviewer's comment that raised the necessity for an elaboration of the IG-decoupling sequence in ^{13}C NMR analysis. Therefore, we have made an appropriate adjustment in our new manuscript.

Relevant Section in the New Manuscript: Line 154 – 157

For a quantitative analysis of the ratio between allyl moieties and the AGU unit in one AC NP, IG-decoupling sequence was performed. By gating off the ^{13}C spin over the acquisition time only, the ^{13}C decoupling sequence reduces the Nuclear Overhauser Effect (NOE) enhancements of carbon atoms.³⁹

Line 154-166: The authors correctly stated that CP-MAS NMR is not quantitative. However, it is hard to follow their use of this technique to quantify the percentage of allylation in the LAC sample. First, in order to compare the relative intensities of ^{13}C -NMR signals across different samples is better to use an internal standard instead of the other cellulose signals. Second, did the authors take the IG NMR spectrum for LAC? What was the result? Third, significant figures of the ratios and percentages need to be double-checked. Fourth, there is a typo in line 166, instead of Table S1 it should say Table 1.

We greatly appreciate and agree with reviewer's concern over the method our group used to approximate % allylation. Therefore, we conducted solid state ^{13}C IG-NMR for LAC specimen separately to obtain its quantitative % allylation. Fig. R3a and R3b show solid state ^{13}C CP-MAS NMR spectra of unmodified CNC, AC, and LAC, and IG-NMR spectra of AC and LAC, respectively.

In the case of AC (Fig. R3a), carbon chemical shifts (δ) of allyl group ($-\text{CH}_2\text{CH}=\text{CH}_2$) were observed at 134.5 ppm (C8), 116.0 ppm (C9), and 29.22 ppm (C7) in the solid state ^{13}C CP-MAS analysis, which agreed with previously reported results.^{1,2} These peaks were also observed in the CP-MAS NMR spectrum of LAC at similar δ : 135.3 ppm (C8), 117.9 pm (C9),

30.69 ppm (C7). Compared to other peaks assigned for allyl groups, those assigned to C7 of AC and LAC lacked clarity due to the newly built up NOE which may be caused by several experimental parameters disproportionally enhancing certain carbon signals and producing non-quantitative spectra.³ The quantitative IG-NMR analyses of the prepared specimen also revealed carbon characteristic peaks of allyl groups at similar δ values ($\delta_{AC,C8} = 134.7$ ppm, $\delta_{AC,C9} = 116.1$ ppm, and $\delta_{AC,C7} = 29.15$ ppm; and at $\delta_{LAC,C8} = 133.6$ ppm, $\delta_{LAC,C9} = 117.2$ ppm, and $\delta_{LAC,C7} = 30.54$ ppm). Our group chose the carbon peak at C1 position of AC/LAC AGU unit as the standard peak (1.00). The integrated carbon peak values assigned to allyl-groups were 0.1363 (C7), 0.9210 (C8), and 1.3199 (C9); and 0.1325 (C7), 0.2159 (C8), and 0.0441 (C9) for AC and LAC, respectively. Since an AGU unit of CNC contains three OH groups that could be modified, a complete allylation or the sum of peak integration across C7, C8, and C9 was considered to be equal to 3. Therefore, % allylation can be obtained by

$$\% \text{ allylation} = \frac{(\text{sum of the obtained peak integrated values of C7, C8, C9})}{3}$$

Using this method, % allylation of AC and LAC are approximately 79 % and 13 %, respectively (Table R1). Unmodified NMR spectra of AC and LAC are provided in Fig. S1 and S2.

References

- (1) Hu, H., You, J., Gan, W., Zhou, J. & Zhang, L. Synthesis of allyl Cellulose in NaOH/urea aqueous solutions and its thiol-ene click reactions. *Polym. Chem.* **6**, 3543-3548 (2015).
- (2) Glaser, R., Hillebrand, R., Wycoff, W., Camasta, C. & Gates, K. S. Near-silence of isothiocyanate carbon in ¹³C NMR spectra: a case study of allyl isothiocyanate. *J. Org. Chem* **80**, 4360-4369 (2015).

- (3) Otte, D. A. L., Borchmann, D. E., Lin, C., Weck, M. & Woerpel, K. A. ^{13}C NMR spectroscopy for the quantitative determination of compound ratios and polymer end groups. *Org. Lett.* **16**, 6, 1566-1569 (2014).

Relevant Section in the New Manuscript: p. 8 Line 160 – 179

In the case of AC (Fig. 2a), carbon chemical shifts (δ) of allyl group ($-\text{CH}_2\text{CH}=\text{CH}_2$) were observed at 134.5 ppm (C8), 116.0 ppm (C9), and 29.22 ppm (C7) in the solid state ^{13}C CP-MAS analysis, which agreed with previously reported results.^{38, 40} These peaks were also observed in the CP-MAS NMR spectrum of LAC at similar δ : 135.3 ppm (C8), 117.9 pm (C9), 30.69 ppm (C7). Compared to other peaks assigned for allyl groups, those assigned to C7 of AC and LAC lacked clarity due to the newly built up NOE which may be caused by several experimental parameters disproportionally enhancing certain carbon signals and producing non-quantitative spectra.⁴¹ The quantitative IG-NMR analyses (Fig. 1b) of the prepared specimen also revealed carbon characteristic peaks of allyl groups at similar δ values ($\delta_{\text{AC,C8}} = 134.7$ ppm, $\delta_{\text{AC,C9}} = 116.1$ ppm, and $\delta_{\text{AC,C7}} = 29.15$ ppm; and at $\delta_{\text{LAC,C8}} = 133.6$ ppm, $\delta_{\text{LAC,C9}} = 117.2$ ppm, and $\delta_{\text{LAC,C7}} = 30.54$ ppm). Using the carbon peak at C1 position of AC/LAC AGU units as the standard peak (1.00), those assigned to allyl-groups were 0.14 (C7_{AC}), 0.92 (C8_{AC}), and 1.3 (C9_{AC}); and 0.13 (C7_{LAC}), 0.22 (C8_{LAC}), and 0.044 (C9_{LAC}). Since an AGU unit of CNC contains three OH groups that could be modified, a complete allylation or the sum of peak integration across C7, C8, and C9 positions was considered to be equal to 3. Therefore, % allylation can be obtained by

$$\% \text{ allylation} = \frac{(\text{sum of the obtained peak integrated values of C7, C8, C9})}{3}$$

Using this method, % allylation of AC and LAC are approximately 79 % and 13 %, respectively

(Table 1).

Relevant Section in the New Manuscript: p. 9 Table 1

Table 2. Quantitative allyl/AGU ratios and the approximated % allylation in AC and LAC were obtained via ^{13}C Inverse-gated Solid NMR (IG-NMR).

	Quantitative Allyl/AGU Ratio	% Allylation
Allyl-Cellulose (AC)	2.4:1	79
Low Allyl-AC (LAC)	0.39:1	13

Relevant Section in the New Manuscript: p. 9 Figure 1

Line 178: The claim needs a reference.

We certainly agree with the reviewer's comment that the general claim about the effects of nanoparticle dispersity to the reaction product. Therefore, the relevant section was edited appropriately.

Relevant Section in the New Manuscript: Lines 189 – 190

Both AC and LAC were suspended in DCPD liquid monomer at five different wt%: 0.5, 1, 2, 4 wt%, prior to being thermally initiated for FP. It is widely understood that the NP dispersity greatly contributes to the characteristics of reaction products.^{44,45}

References

- (44) Naito, M., Yokoyama, T., Hosokawa, K. & Nogi, K. Chapter 3 - Characteristics and Behavior of Nanoparticles and Its Dispersion Systems. in *Nanoparticle Technology Handbook 3*, 109-168 (Elsevier, 2018).
- (45) Joudeh, N. & Linke, D. Nanoparticle classification, physicochemical properties, characterization, and applications: a comprehensive review for biologists. *Journal of Nanobiotechnology* **20**:262 (2022).

Line 181: It is unclear why the authors studied the agglomeration in aqueous media instead of the monomer.

We appreciate the reviewer's concern as for the reason why the agglomeration of AC was done in aqueous media (i.e., water), not in liquid monomer. In our manuscript, we mentioned that DCPD was solid at room temperature and becomes liquidous at an elevated temperature (50°C). At a depreciated temperature, the monomer turned back into solid. These

conditions made difficult for a TEM analysis confirming the dispersity or agglomeration of AC in DCPD monomer to be conducted. Since the objective of the TEM analysis was to observe the dispersity of AC due to its surface modification and size, we had dispersed our material in water which has a relatively similar low-viscosity (~1 cP) compared to that of the monomer previous work⁴ had reported (~1.5 cP).

References

- (4) Robertson, I. D. *et al.* Rapid energy-efficient manufacturing of polymers and composites via frontal polymerization. *Nature* **557**, 223-227 (2018).

Line 193-197: If the authors started with the same mass in all the experiments, why did they observe a decrease in the mass after polymerization? It is not clear why by using a higher percentage of AC the authors obtained foams with lower mass.

We incredibly appreciate the reviewer's critical comment pointing out the mass reduction that was observed from our result. From the current state of study, the reason for mass reduction could not be fully elucidated. However, from Tables S1 and S2, it may be observed that the density of the produced DCPD/AC and -LAC foams generally decreased for increasing the wt% of AC and LAC, respectively. According to a previous work⁵, the more extensive foaming process (i.e., higher interconnectivity and increased porosity) by the addition of reactive reactants may be related to the decrease in density. Since with the increasing % allylation (LAC, 13%, to AC, 79%) and AC wt% seemed to have steadily decreased the density of the reaction product, we presumed this effect resulted from the more aggressive foaming process which may be observed by the increasing pore sizes and % porosity from the μ -CT analysis.

References

- (5) Silverstein, M. S. Emulsion-templated polymers: contemporary contemplations. *Polymer* **126**, 261 – 282 (2017)

Line 200: The authors claim the directional anisotropy is maintained. However, there are no calculations that support this claim.

We greatly appreciate the reviewer's suggestion to obtain quantitative data that showed directional anisotropy (DA) of the DCPD/AC and -LAC foams. DA was calculated by analyzing μ -CT images of the coronal planes on the Bruker CTAn software. Briefly, the software calculated through the mean intercept length (MIL) analysis in which a grid of lines was sent through the binarized image, and the length of the test lines were divided by the number of intercepts (eigenvalues) in a certain volume. Due to its volume-specific nature, we created a rectangular ROI (750 pixels x 800 pixels) to measure DA over a specific volume and analyzed 550 "slices" of μ -CT images.

As shown in Figure 3e, for increasing AC wt%, a relative decrease in DA was observed, but within the range of anisotropic DA that was previously reported by D. M. Alzate-Sanchez *et al.*²⁴. On the other hand, some DCPD/LAC samples displayed comparable DA values to those obtained for DCPD/AC. From Fig. 3a and Table 2, it can be seen that with increasing AC wt%, pores became larger and the interconnectivity between the formed pores became increasingly apparent. On the other hand, as shown in Fig. 3c, DCPD/LAC foams exhibited poor formation of pores thus displayed significantly lower interconnectivity between the pores. In this sense, our group postulated that differentiation of individual but connected pores were not made by the software. Therefore, like it is shown in Fig. S7, we speculated that the slight decrease in anisotropy was due to the pore interconnectivity by the increasing pore sizes. Therefore, we sought to measure the % porosity along the coronal planes to support our claim.

As shown in Fig. 3e, a clear trend of increasing % porosity and mildly decreasing DA was observed with increasing AC wt%. Unlike DCPD/AC foams, DCPD/LAC foams exhibited significantly lower porosity from which DA stayed relatively constant across the examined samples. By these findings, it was understood that with increasing AC wt%, pore diameters of the formed millichannel increased, while maintaining their directional anisotropy. These results were discussed in the relevant section: page 10, lines 187 – 227; of the newly edited manuscript.

Relevant Section in the New Manuscript: Figure S7

Figure S7. Degree of anisotropy (DA) were measured using the Bruker CTAn analyzer. Briefly, DA values are obtained as eigenvalues by the mean intercept length (MIL) calculation. With increasing AC wt%, the formed pore walls were interconnected to each other by which are indicated as one large pore with various directions on the software.

Relevant Section in the New Manuscript: **Figure 3**

Figure 3. (a) μ -CT images obtained for DCPD/AC foams and (b) the measured average pore diameters (mm) were plotted against % frequency. Similarly, μ -CT images of (c) DCPD/LAC foams and their (d) average pore diameters (mm) were plotted against % frequency plot. (e) % porosity and DA of DCPD/AC and -LAC foams were obtained using the Bruker CTAn analyzer.

Relevant Section in the New Manuscript: **Table 2**

Table 2. Average diameters of DCPD/AC and DCPD/LAC foams, and their standard deviation values are shown.

Avg. Diameter (mm)	0.5 wt%	1 wt%	2 wt%	4 wt%
AC	0.64 ± 0.4	1.1 ± 0.8	1.2 ± 0.7	1.6 ± 1
LAC	1.4 ± 0.5	1.9 ± 1	0.76 ± 0.4	1.2 ± 0.7

Line 202-208: The authors suggest LAC performs worse than AC in forming polymeric solid foams. However, this claim is based solely on the comparison of images 4a and 4b. Authors need to calculate alignment, average diameter, and distribution to support this claim.

We greatly appreciate the reviewer’s insightful suggestion to present additional quantitative data that proved our claim of the foam production using LAC. In the newly prepared Fig. 3 and Table 2, we have included DA, average diameters, and pore size distributions of the produced DCPD/LAC foam.

Relevant Section in the New Manuscript: **Figure 3, Table 2**

Line 224: It is not clear the use of the arrow to indicate the direction of propagation in Figures 4a and 4b. The figures suggest the authors are showing not just the parallel section but also the transversal one.

We appreciate the reviewer’s concern over the overall readability of the data presented in Figure 4. Following the reviewer’s advice, we have made an appropriate correction in the newly prepared Figure 3.

Relevant Section in the New Manuscript: **Figure 3, Table 2**

Line 228: Double-check the significant figures in Table 2 and reconsider the importance of the inclusion of R2.

We certainly agree with the reviewer's comment regarding the necessity for correct significant figures in reporting the average diameters of the pores formed in the DCPD/AC foams. Moreover, with the appropriate addition made in Figure 3d, we have also included average diameters of the pores formed in DCPD/LAC foams in Table 2.

Relevant Section in the New Manuscript: Figure 3, Table 2

Line 231-232: Reconsider rephrasing for more clarity.

We greatly appreciate the reviewer's suggestion to provide more clarity in elaborating our claim about the phase separation-mediated foaming mechanism presented in this study. Based on the reviewer's suggestion, we have made appropriate changes to the relevant section in our newly written manuscript.

Relevant Section in the New Manuscript: Lines 235 – 258

In the conventional methods of producing porous DCPD solids, the monomer (minority component) is either dissolved in a solvent (i.e., chemically induced phase separation, CIPS) or mixed with an immiscible liquid (80 – 99 vol%) (i.e., high internal phase emulsion, HIPE) from which the polymerization of a continuous DCPD phase triggers CIPS or HIPE within the biphasic polymerization system *in situ*.^{24,36,47} Once DCPD is polymerized, the non-polymeric phases have to be removed to reveal the formed macroporous structures. On the other hand, the current study may combine CIPS and HIPE in which both the majority (DCPD,

96 – 99.5wt%) and the minority (AC, 0.5 – 4wt%) components were simultaneously polymerized into continuous phases that also created a macroporous architecture in a one-pot FP system.

To support the claim that phase separation drove the production of DCPD/AC foams, our group first hypothesized that no new C=C bonds between the two reactants would have formed in discrete metathesis polymerizations of DCPD and AC. Therefore, solid state ^{13}C CP-MAS NMR analysis of the produced DCPD/AC foams was conducted to examine any changes (e.g., shifts in δ or peak intensities) to the C=C bond peaks at approximately 130 ppm (C8, C10) in parallel with increasing AC wt% (Figure S8). Despite the increase in AC wt%, no significant δ shifts or peak intensity changes were observed from the ^{13}C CP-MAS NMR spectra of the prepared foams (0 – 4 wt% AC). On the other hand, the X-ray diffraction (XRD) analysis of the same specimens displayed an increasing trend of intensity pertaining to the cellulose (004) plane characteristic peak³⁷ at approximately $2\theta \approx 31.7^\circ$ with the increment of AC wt% (Figure S9). Collectively, these results may suggest that the C=C moieties of ACs did not participate in the ROMP of DCPD while being present in the one-pot production of FP-mediated DCPD/AC foams.

References

- (24) Alzate-Sanchez, D. M. *et al.* Anisotropic foams via frontal polymerization. *Adv. Mater.* **34**, 2105821 (2022).
- (36) Kovačič, S. & Slugovc C. Ring-opening metathesis polymerisation derived poly(dicyclopentadiene) based materials. *Mater. Chem. Front.* **4**, 2235-2255 (2020).
- (37) Hu, Z., Berry, R. M., Pelton, R. & Cranston, E. D. One-pot water-based hydrophobic surface modification of cellulose nanocrystal using plant polyphenols. *ACS Sustainable Chem. Eng.* **5**, 5018–5026 (2017).

(47) Zhang, T., Sanguramath, R. A., Israel, S. & Silverstein, M.S. Emulsion templating: porous polymers and beyond. *Macromolecules* **52**, 5445-5479 (2019).

Relevant Section in the New Manuscript: Figure S8, S9

Figure S8. ¹³C CP-MAS solid NMR spectra of unmodified CNC, AC, LAC, neat DCPD, and DCPD/AC foams of varying AC wt% are displayed. Although increasing wt% of AC were introduced to DCPD, no chemical shifts or intensity changes at around 130 ppm (C=C bond) were observed compared to the neat DCPD peak.

Figure S9. (a – b) The X-ray diffraction (XRD) patterns of unmodified CNC, AC, neat DCPD, and DCPD/0.5 – 4wt% AC foams are shown. With increasing AC wt%, peak intensity pertaining to the cellulose (004) plane became increasingly apparent, suggesting the unlikelihood of ACs being interacted with DCPD undergoing ROMP while being present in the same reaction pot.

Line 231-233: The claim about the no crosslinking between DCPD and AC is hard to support by ^{13}C -NMR because at these concentrations the AC signals are not observed in Figure S5. Therefore, it is highly unlikely to observe a signal corresponding to cross-metathesis in the pDCPD signal around 130 ppm.

We greatly appreciate the reviewer's critical point made over the originally presented solid state ^{13}C CP-MAS NMR spectra of CNC, AC, and DCPD/AC foams in the explanation of the phase-separation mediate foaming mechanism.

In the case of the CP-MAS NMR analyses of neat DCPD and DCPD/AC foams, we were looking not only looking for potential carbon shifts (δ) but also the changes in the peak intensity at 130 ppm as it corresponded to C=C bonds from the peak assigning of AC, LAC, and DCPD. We hypothesized that if there were chemical interactions between the allyl groups of AC and the crosslinking C=C groups of DCPD, any chemical signs induced by this effect would be observed through the solid ^{13}C CP-MAS NMR analysis. Without significant changes to those parameters being observed, we conducted an X-ray diffraction (XRD) analysis using the same specimens and observed a characteristic peak of the cellulose (004) plane at around $2\theta \approx 31.7^\circ$. Collectively, we presumed that ACs may not have participated in the ROMP

reaction of DCPD while being present in the one-pot production of FP-mediated DCPD/AC foams.

These new explanations were included in the newly edited version of our manuscript at their relevant sections.

Relevant Section in the New Manuscript: Lines 235 – 258

Relevant Section in the New Manuscript: Figures S8, S9

Line 237-239: The authors suggest that bubbles are forming from a phase separation phenomenon. It would be important that the authors expand this concept. Where are the bubbles coming from? Are they formed from dissolved gases? Why phase separation produces bubbles in this system?

We greatly appreciate the reviewer's suggestion to include the elaboration of phase separation and the subsequent bubble formation.

In the edited section of our manuscript, we have compared the already known methods of producing porous DCPD: high internal phase emulsion (HIPE) and chemically induced phase separation (CIPS), to our method. We have also investigated the feasibility of other reaction parameters such as the initiator/reactant degradation, solvent boiling, and dissolution of gas into the liquid monomer/AC mixtures that might have played a fundamental role in the bubble formation. After deducing the factors that were unlikely to have induced foaming of DCPD/AC or -LAC mixtures, our group explored the more traditional foaming process that was widely accepted as the underlying mechanism in the foam production *via* the HIPE method: bubbles were formed by the evacuation of the individually dispersed droplets during the formation of a new continuous internal phase. Under the presumption that the C=C moiety on ACs were undergoing CM, we sought to trace the remarks of crosslinking by AC allyl groups

from the produced foams, the reaction kinetics (e.g., front velocity, initiation time, and T_{Max}). The more detailed explanation related to the bubble formation by phase separation may be found in the following articles.

Relevant Section in the New Manuscript: Lines 235 – 387, Figures 4, 5, Table 3

In the conventional methods of producing porous DCPD solids, the monomer (minority component) is either dissolved in a solvent (i.e., chemically induced phase separation, CIPS) or mixed with an immiscible liquid (80 – 99 vol%) (i.e., high internal phase emulsion, HIPE) from which the polymerization of a continuous DCPD phase triggers CIPS or HIPE within the biphasic polymerization system *in situ*.^{24,36,47} Once DCPD is polymerized, the non-polymeric phases have to be removed to reveal the formed macroporous structures. On the other hand, the current study may combine CIPS and HIPE in which both the majority (DCPD, 96 – 99.5wt%) and the minority (AC, 0.5 – 4wt%) components were simultaneously polymerized into continuous phases that also created a macroporous architecture in a one-pot FP system.

To support the claim that phase separation drove the production of DCPD/AC foams, our group first hypothesized that no new C=C bonds between the two reactants would have formed in discrete metathesis polymerizations of DCPD and AC. Therefore, solid state ^{13}C CP-MAS NMR analysis of the produced DCPD/AC foams was conducted to examine any changes (e.g., shifts in δ or peak intensities) to the C=C bond peaks at approximately 130 ppm (C8, C10) in parallel with increasing AC wt% (Figure S8). Despite the increase in AC wt%, no significant δ shifts or peak intensity changes were observed from the ^{13}C CP-MAS NMR spectra of the prepared foams (0 – 4 wt% AC). On the other hand, the X-ray diffraction (XRD) analysis of the same specimens displayed an increasing trend of intensity pertaining to the cellulose (004) plane characteristic peak³⁷ at approximately $2\theta \approx 31.7^\circ$ with the increment of AC wt% (Figure S9). Collectively, these results may suggest that the C=C moieties of ACs did not participate in

the ROMP of DCPD while being present in the one-pot production of FP-mediated DCPD/AC foams.

The voids or bubbles formed in the produced foams may also corroborate our hypothesis that FP-driven phase separation generated the cellular DCPD/AC solids. According to previous studies, these bubbles were generated by the evacuation of the dispersed individual “droplets” during the formation of a new continuous internal phase⁴⁷, the elimination of the dissolved gas or water in monomer⁴⁸, and during solvent boiling or thermal decomposition of initiators⁴⁹. In the case of the current study, the N₂-purged reaction environment, and the degradation of either DCPD, AC, or LAC seemed less likely to have induced the formation of bubbles. First, the reaction of DCPD/0.5-1 wt% CTD, which were performed in the same reaction environment as those of DCPD/AC and -LAC, did not exhibit any pore formation. Moreover, the thermogravimetric analysis (TGA) of AC and LAC (Figure S10) revealed their robust thermal stability at temperatures (~260 °C) that were significantly higher than the measured maximum reaction temperatures of this study (~150 °C, shown in Fig. 6c), deviating from the likelihood of decomposition of reactants to induce bubble formation during FP.

Based on these results, we hypothesized that the bubbles observed in DCPD/AC and -LAC foams were the outcomes of more than one chemical reaction (ROMP of DCPD) that was present in a FP reaction. Specifically, we presumed that the foaming mechanism in this study involved the generation of an interconnected internal phase by the CM reaction of the AC C=C moieties while DCPD was undergoing a FROMP. Therefore, the amount of allyl moieties of the reactant may be correlated to the formation and the structures of pores in the produced DCPD/AC and -LAC foams. As observed from the μ -CT images, all DCPD/AC monoliths and those of DCPD/LAC at higher LAC wt% globally displayed robust formations of pores that were arranged in a splayed pattern (Figure 3a, 3c). Previously, A. J. Pojman reported a formation of similar bubble pattern that were indicative to a macroscopic phase separation

induced by the individual crosslinking of two components in one system.⁵⁰ The field-emission scanning electron microscopy (FE-SEM) images of DCPD/AC foams further displayed the effects of individual crosslinking of the monomer and AC to the construct of subsequently polymerized millichannel microstructures during FP. Compared to the cross-section surface of neat DCPD solid, as shown in Fig. 4a, millichannels walls of DCPD/AC foams exhibited smooth curvatures that may be representative to the pDCPD surfaces nucleated by a simultaneous reaction of ROMP and CM (Figure 5b-e). These surface morphologies were also observed in previously reported frontally produced DCPD solids that leveraged a synchronized reaction of binary reactant systems such as ROMP of DCPD/1,5-cyclooctadiene²², and ROMP of DCPD/decomposition of cyclohexane²⁴. The advent of spin mode, shown as cross-sectional image of a spiral pattern from the μ -CT and digital optical microscopy (OM) images of lower AC wt% (0.5-1wt%) (Figure 3a, 4b, S11a-b), was another feature of FP-driven phase separation that may have arisen from distinct chemical crosslinking of two C=C containing reactants. Briefly, the spin mode of FP results from the temperature perturbation results in an unstable two-dimensional front that simultaneously moves to its perpendicular directions.⁵⁰ According to previous studies, this periodic mode was influenced by the initial reaction temperature, the geometry of reactor in which FP occurs, and the varying crosslinking degree within the reaction.⁵⁰⁻⁵² Consequently, we presumed that the parameter leading to the occurrence of spin mode was also related to the production of cellular solids *via* FP. Since the initial experimental temperature was constant at room temperature (25°C), we first investigated the effect of changing the glass tube diameter. DCPD/0.5wt% AC mixture was frontally polymerized separately in a glass vial and a borosilicate capillary tube with the diameter (d) of 25 mm and 1.5 mm, respectively (Figure S11a, S11b). Due to the differences in the thickness of the produced monoliths, foam millichannels that were reproduced in the capillary tube and the glass vial were observed through the digital OM and the FE-SEM, respectively. Nevertheless,

both results clearly displayed millichannels formed *via* single head spin mode during DCPD/AC FP with relatively consistent periodicity ($230 \pm 32 \mu\text{m}$ for $d = 1.5 \text{ mm}$; $170 \pm 41 \mu\text{m}$ for $d = 25 \text{ mm}$). These results supported the unlikeliness of the reactor geometry of being the major contributor to have induced a spin mode during the frontal production DCPD/AC foams.

Figure 4. Field emission scanning electron microscope (FE-SEM) images obtained for (a) neat DCPD solid, (b) DCPD/0.5wt% AC, (c) DCPD/1wt% AC, (d) DCPD/2wt% AC, and (e) DCPD/4wt% AC at x100 – 500 magnification.

Interestingly, however, millichannels became increasingly wider by being merged with one another when AC wt% increased (Figure S11b–e). This suggested that the amount of C=C moieties being reacted during FP, or the changes in crosslinking degree in effect to the varying amount of C=C groups, affected the construct of DCPD/AC foams. Given that FP is an exothermic reaction, we postulated that an addition of allylated CNCs could alter the reaction dynamics of a conventional FROMP of DCPD by triggering a simultaneous reaction of ROMP and CM of DCPD monomer and AC, respectively. To investigate the effects of increasing C=C

moiety to the dynamics of a GC2-mediated FP reaction, we sought to measure the kinetics at the reaction front of the bulk polymerization of DCPD/AC mixtures^{16,53,54}. Therefore, we observed the changes in front velocity (v_f) and maximum reaction temperatures (T_{Max}) of the prepared monomer-AC mixtures during FP. Front displacements (x) were first measured by monitoring the moving front positions across the test tubes that were pre-labeled at 1 cm intervals. The obtained x values were plotted against time (t) to calculate v_f of each reaction. T_{Max} values were obtained by using T-type thermocouples that were submerged to the bottom of the glass tubes containing the liquid mixtures, from where a thermal stimulus was applied and initiated FP. As the reactions were initiated, the propagating fronts started curing the liquid DCPD/AC mixture along the direction of the transferring thermal energy (Figure 5a). The x vs. t plot showed stable, linearly increasing graphs of the prepared DCPD/AC specimens, indicating pure FPs (Figure 5b). As shown in Table 3, v_f steadily increased from 0.89 ± 0.2 mm s^{-1} to 1.7 ± 0.1 mm s^{-1} as AC wt% increased up to 2wt%, then slightly decreased to 1.1 ± 0.1 mm s^{-1} when it reached 4wt%. The relative decrease in v_f in the case of 4wt% AC may have resulted from the mild interruption in the chain reaction during CM due to the increased amount of non-allylated hydroxyl groups on CNC surfaces, since % allylation in AC did not yield 100% from our result.⁵⁵ Nevertheless, similar trend of increasing v_f with increasing C=C moieties was reported in a number of previously studied FP systems whose feature was attributed to the increased crosslinking degree.^{56,57} Experimentally, this phenomenon may be better understood by comparing the reaction velocities of DCPD/AC, -LAC, and -CTD. Figure S12a depicts the drastic differences in v_f in relations to the presence of allyl groups, % allylation of CNCs, and their wt% in FP reactions. The effect of allylating CNCs in FP can be clearly observed, as compared to neat DCPD, FP of DCPD/AC and -LAC displayed increased v_f while DCPD/CTD samples exhibited significant decrease in v_f with increasing CTD wt%., presumably due to the hindrance of an uninterrupted reaction by the non-allylic moieties. The impact of % allylation

pertaining to the reaction dynamics can also be realized. While FP containing ACs generally showed increasing trend of v_f , those incorporating LACs revealed negligible differences between the measured v_f values. This phenomenon appears clearer when comparing the smallest wt% of AC, LAC, and CTD in FP of DCPD (Figure S12b). In this sense, the effect of incorporating highly reactive olefin metathesis group like allyl-moiety into FP of DCPD was translated into an overall increase in v_f that exhibited further rise upon the compositional increment of reactive C=C bonds. Figure 6c shows a reaction temperature profile of neat DCPD and DCPD/0.5 – 4 wt% AC for which temperature values were recorded at every two seconds for the duration of the FP reaction. The graphs were characterized by sharp temperature spikes (i.e., T_{Max}) that were observed shortly after the reaction was thermally initiated. Without the application of local stimulus, the reaction temperature remained rather constant at around 21 °C. The flatter regions of the graphs indicate the reaction initiation time ($t_{initiation}$) or the amount of time the initiation zone was heated with a soldering iron before front propagations were observed. With increasing AC wt%, the $t_{initiation}$ increased with the exception of 2wt% AC from which the quickest reaction initiation was observed. Nonetheless, the general trend of increasing $t_{initiation}$ was also observed from the previously reported FP systems with secondary reactive components that slightly lowered the reactivity of reactions.^{24,58} Especially, when micro-/nanoparticles were added into these systems, as heat sinks, they absorbed the released heat and also decreased T_{Max} values of the reactions.^{21,56,59} On the other hand, our system exhibited small differences between T_{Max} values of neat DCPD and the prepared DCPD/AC formulations at around 151 °C (Table 3). We first investigated the size of ACs as a possible factor to have induced such effect to T_{Max} . However, the dimension ($w \times l$) of AC was approximately $14 \pm 3 \times 88 \pm 13$ nm (aspect ratio $\sim 6.6 \pm 2$) from the TEM analysis (Figure S6), which its width (w) was similar to the sizes of spherical SiO₂ NPs in FP systems that were previous studied by S. P. Davtyan *et al.*⁵⁹ (10 nm) and S. Chen *et al.*⁵⁶ (20 nm). Meanwhile, a

recent study by L. M. Dean *et al.*¹⁹ exhibited the effect of increasing aspect ratios of carbon-based NPs to enhancing the kinetics of an FP system such as thermal conductivity, T_{Max} , initiation, and v_f . Based on these previous findings, we postulate that the thermal insulating nature of AC NPs and their relatively high aspect ratio offset each other's effects, leading into the small shifts in T_{Max} . As for the rather increasing standard deviation of T_{Max} values, we presume that the stochastic formation of bubbles during FP of DCPD/AC made it challenging for the thermocouple sensor to not only detect the sole instantaneous T_{Max} of the dynamic pDCPD front but also to differentiate the temperature of pDCPD and the simultaneously generated air-filled voids (Figure S13a). This phenomenon may be further demonstrated from the μ -CT images that visualized the positions of the thermocouple hot junctions being inside the pores of the DCPD/AC foams (Figure S13b-e).

Figure 5. FP of DCPD/AC was recorded using an IR- and a digital camera. (a) For increasing AC wt%, the reaction lasted for shorter amounts of time. (b) Front velocities (v_f) of each reaction were obtained by plotting front displacement (x) against time (t). (c) The temperature vs. time plots exhibited negligible changes in T_{Max} for increasing AC wt%. On the other hand, a general trend of increasing reaction initiation time ($t_{initiation}$) was observed with increasing AC wt%. (d) Reaction kinetics of 2wt% AC was highlighted as it exhibited the most rapid v_f and $t_{initiation}$ amongst the tested specimens.

Table 3. Front velocity (v_f), maximum temperature (T_{Max}), and initiation time ($t_{initiatoin}$) values were obtained during frontal polymerization of the DCPD/AC samples.

	0 wt%	0.5 wt%	1 wt%	2 wt%	4 wt%
v_f (mm s ⁻¹)	0.89 ± 0.2	1.2 ± 0.05	1.3 ± 0.03	1.7 ± 0.1	1.1 ± 0.1
T_{Max} (°C)	151 ± 0.20	153 ± 5.4	150 ± 9.3	153 ± 17	151 ± 15
$t_{initiation}$ (s)	22 ± 6	60 ± 1	106 ± 4	36 ± 8	110 ± 8

Figure S13. (a) A schematic is shown in support of our presumption that the stochastically growing nature of bubbles led to the large standard deviation due to the incomplete contact between the thermocouple hot junction and the polymerizing monomer. (b) μ -CT images of the polymerized foams showed that the hot junctions of the thermocouples were sometimes located inside the pores (dark areas) or in contact with the polymer (grey area).

Relevant Section in the New Manuscript: Lines 235 – 387, Figures 4, 5, Table 3

Figure S10. Thermogravimetric analysis of (a) unmodified CNC, (b) CTD, (c) AC, and (d) LAC. These reactants displayed robust thermal stability at temperatures that were significantly higher than the maximum reaction temperatures observed in this study.

Relevant Section in the New Manuscript: Figure S8, S9

Relevant Section in the New Manuscript: Figure S11

Figure S11. (a) An image of a cross section of DCPD/0.5wt% AC monolith produced in a $d = 25$ mm glass vial obtained from the FE-SEM analysis. (b) Digital optical microscope image ($\times 0.63$) of the DCPD/0.5wt% AC mixture cured in a $d = 1.5$ mm capillary tube. (c – e) Digital optical microscope images ($\times 0.63$) of DCPD/1-4wt% AC showed that with increasing AC wt%, the formed millichannels merged with each other thereby forming wider pores.

Line 239-250: The authors claim that the wrinkles observed in the foams are caused by the polymerization rate difference between ROMP and CM. However, as the authors stated, this behavior had been observed in another system where there is only one type of reaction (ROMP). Therefore, the claim can't be supported by the experiment.

We certainly agree with the reviewer's comment that pointed out the ambiguity of the claim we have made in our original manuscript. To improve clarity of the manuscript, we have edited out the addressed portion of the manuscript. Meanwhile, in the case of the smooth curvature that was mentioned in the original manuscript, we have included additional references to support our claim that the smooth nucleated surface did result from a simultaneous reaction of two reactants in one reaction system.

Compared to the cross-section surface of neat DCPD solid, as shown in Fig. 4a, millichannels walls of DCPD/AC foams exhibited smooth curvatures that may be representative to the pDCPD surfaces nucleated by a simultaneous reaction of ROMP and CM (Figure 4b-e). These surface morphologies were also observed in previously reported frontally produced DCPD solids that leveraged a synchronized reaction of binary reactant systems such as ROMP of DCPD/1,5-cyclooctadiene²², and ROMP of DCPD/decomposition of cyclohexane²⁴.

Line 248-267: The authors discussed the observation of spin mode in formulations with low content of AC. However, no connection of the spin mode with the hypothesis of the manuscript was provided. It is hard to understand the reason why this observation was included in the manuscript. Moreover, Figures 4a, 4b, and S3a-b are not related to the study of the spin mode. Why was spin mode not observed in the other formulations? Why were two different imaging techniques used to compare Figure S6?

We greatly appreciate the reviewer's concern over the discontinuity in the proposed hypothesis and the observation of spin mode from our original manuscript. Based on the suggestion that the reviewer had given us in the previous comments, we have included more corroborated explanation as to how the discovery of spin mode from the lower AC wt% foams intrigued us to further explore the possibility of varying crosslinking degree and determine the heart of the foaming mechanism.

Moreover, in Fig. S11, we have shown the formed voids in the DCPD/AC foams that were produced in 1.5 mm borosilicate capillary tubes. Here, with increasing AC wt%, the walls spiral-like voids started to merge with each other that consequently produced larger pores. From this, it was understood that the apparent sign of spin mode was not observed from higher AC wt% due to the increased pore sizes by the merging of voids that were produced by the spin

mode of FP. Furthermore, as reasoned in our newly prepared manuscript, FP of DCPD/0.5 wt% AC was performed in two different reactors during the investigation of the leading cause of the spin mode. While the inner void structure was visible through the digital OM for thinner (1.5 mm in thickness) foam specimens, those of the larger (25 mm in thickness) specimens required for additional preparation of the sample to visualize its void microstructure.

Relevant Section in the New Manuscript: Lines 235 – 387, Figures 4, 5, Table 3

Line 274: The authors should include the images of other formulations together with Figure S6c to support the increase in the multichannel size.

We appreciate the reviewer's suggestion to include images of other AC formulations that were frontally polymerized in borosilicate capillary tubes with diameters of 1.5 mm. We have provided the relevant images in Fig. S11.

Relevant Section in the New Manuscript: Figure S11

Line 282: The use of an IR camera to measure maximum temperature is not accurate because glass absorbs IR radiation.

We most definitely agree with the reviewer that using an IR camera from a distance for T_{\max} measurement was not the most accurate method. Thus, we obtained temperature values for every 2 seconds intervals from the inside of the sample using commercially available T-type thermocouples from Omega Engineering. With these new set of data, we provide a more reliable set of T_{\max} values of neat DCPD (AC 0 wt%), and DCPD/AC 0.5, 1, 2, 4 wt%. Briefly, we observed little differences between the T_{\max} values at around 151 °C. This observation was in contrast to previously reported particle-mediated FP systems where NPs, as thermal sinks, partially absorbed the heat released during the reaction.^{1,2,3} The size (width) of NPs seemed to

induce negligible effect to T_{Max} as the size AC ($w \times l$), from our TEM result, was approximately 14 x 88 nm which fell in between the sizes of spherical SiO₂ NPs of 20 nm and 10 nm reported by S. Chen *et al*¹ and S. P. Davtyan *et al*², respectively. Meanwhile, a recent work exhibiting increased T_{Max} for increasing carbon nanotube (CNT) wt% in a DCPD FP system was reported by L. M. Dean *et al*⁴. According to their work, this phenomenon was attributed to the enhanced thermal conductivity during FP induced by the high aspect ratio of CNTs, as well as carbon-based NPs' robust intrinsic photothermal conversion efficiencies.⁴ Based on these previous studies, it may be understood that AC's nonconductivity and high aspect ratio offset each other thus resulted in the small T_{Max} shifts compared to that of neat DCPD.

These findings and the related explanations have been edited into relevant sections in the new manuscript.

References

- (1) Chen, S., Sui, J., Chen, L. & Pojman J. A. Polyurethane-Nanosilica Hybrid Nanocomposites Synthesized by Frontal Polymerization. *J. Polym. Sci. A Polym. Chem.*, **43**, 1670-1680 (2005).
- (2) Davtyan, S. P., Berlin, A. A., Shik, K., Tonoyan, A. O. & Rogovina, S. Z. Polymer Nanocomposites with a Uniform Distribution of Nanoparticles in a Polymer Matrix Synthesized by Frontal Polymerization Technique. *Nanotechnol Russia*. **4**, 489-498 (2009).
- (3) Gao, Y. *et al*. Controllable Frontal polymerization and spontaneous patterning enabled by phase-changing particles. *Small* **17**, 2102217 (2021).
- (4) Dean, L. M., Ravindra, A., Guo, A. X., Yourdkhani, M. & Sottos, N. R. Photothermal initiation of frontal polymerization using carbon nanoparticles. *ACS Appl. Polym. Mater.* **2**, 4690-4696 (2020).

Line 290-298: Authors suggest the increase in frontal velocity is due to an increase in the amount of monomers containing C=C moieties. To support this claim authors need to demonstrate that the amount of C=C coming from the AC is higher than the amount of replaced C=C from DCPD. Additionally, this suggestion does not explain why the 4 wt% has a lower frontal velocity. Claim about allyl-moieties being more reactive than cyclopentene is misleading because dicyclopentadiene contains both high strain and low strain cyclic alkenes. Therefore, a discussion about the low reactivity of one particular moiety cannot be done separately.

We greatly appreciate the reviewer's suggestion to retest the FP reaction of DCPD/4wt% AC. First, in Fig. S12, the differences in v_f values in effect to the amount of C=C moieties was highlighted by the FP reactions of DCPD/AC, -LAC, and -CTD. In the case of DCPD/4wt%, the newly obtained v_f may be observed from Fig. 5. Briefly, v_f steadily increased from $0.89 \pm 0.2 \text{ mm s}^{-1}$ to $1.7 \pm 0.1 \text{ mm s}^{-1}$ as AC wt% increased up to 2wt%, then slightly decreased to 1.1

$\pm 0.1 \text{ mm s}^{-1}$ when it reached 4wt%. The relative decrease in v_f in the case of 4wt% AC may have resulted from the mild interruption in the chain reaction during CM due to the increased amount of non-allylated hydroxyl groups on CNC surfaces, since % allylation in AC did not yield 100% from our result.⁵⁵

(55) Groce, B. R., Gary, D. P., Cantrell, J. K. & Pojman A. J. Front velocity dependence on vinyl ether and initiator concentration in radical-induced cationic frontal polymerization of epoxies. *J Polym Sci.* **59**, 1678-1685 (2021).

Relevant Section in the New Manuscript: Figure S12

Relevant Section in the New Manuscript: Figure 5

Line 299-301: Discussion about the relation between monomer consumption and frontal velocity is hard to follow and does not explain the other results.

We certainly agree with the reviewer that the previously reported claim lacked coherency and evidence. In the case of the FP reaction of DCPD/4wt% AC, we presumed that the slight decrease in v_f was resulted from the mild interruption in the chain reaction during CM due to the relative increase in the amount of unreacted hydroxyl groups on the CNC surfaces as % allylation was approximately 79 %.

Relevant Section in the New Manuscript: Figure 5

Line 313: Values in the table need standard deviation

We greatly appreciate the reviewer’s thoughtful suggestion to provide more validity to the reported reaction dynamics values. The appropriate changes were made in the newly prepared manuscript (Table 3).

As for the rather increasing standard deviation of T_{Max} values, we presume that the stochastic formation of bubbles during FP of DCPD/AC made it challenging for the thermocouple sensor to not only detect the sole instantaneous T_{Max} of the dynamic pDCPD front but also to differentiate the temperature of pDCPD and the simultaneously generated air-filled voids (Figure S13a). This phenomenon may be further demonstrated from the μ -CT images that visualized the positions of the thermocouple hot junctions being inside the pores of the DCPD/AC foams (Figure S13b-e).

Relevant Section in the New Manuscript: Table 3

Table 3. Front velocity (v_f), maximum temperature (T_{Max}), and initiation time ($t_{initiatoin}$) values were obtained during frontal polymerization of the DCPD/AC samples.

	0 wt%	0.5 wt%	1 wt%	2 wt%	4 wt%
v_f (mm s⁻¹)	0.89 ± 0.2	1.2 ± 0.05	1.3 ± 0.03	1.7 ± 0.1	1.1 ± 0.1
T_{Max} (°C)	151 ± 0.20	153 ± 5.4	150 ± 9.3	153 ± 17	151 ± 15
$t_{initiation}$ (s)	22 ± 6	60 ± 1	106 ± 4	36 ± 8	110 ± 8

Relevant Section in the New Manuscript: Figure S13

Line 336: What are the intensities the authors are integrating?

We greatly appreciate the reviewer's thoughtful comment. Based on the reviewer's suggestion, we have made an appropriate change to our new manuscript that described the transformation of 2D SAXS/WAXS patterns into 1D azimuthal integrated patterns.

Relevant Section in the New Manuscript: Lines 447 – 450

The obtained 2D scattering images scans were transformed into one-dimensional (1D) scattering patterns using the azimuthal integration method in which the scattered light signals were averaged along the concentric circle around the incident beam by the wavevector (q) or the azimuth angle ($\psi = 360^\circ$).^{67,68}

Line 337-338: The claim about the increase in anisotropy by means of crystallinity cannot be supported by Figure 7a. There are no calculations included about the crystallinity and how it changes by increasing the amount of AC. Additionally, LAC samples do not show any peaks that allow the determination of anisotropy.

We greatly appreciate the critical points raised by the reviewer. Based on the reviewer's suggestion, we have made an appropriate change to our new manuscript that described the transformation of 2D SAXS/WAXS patterns into 1D azimuthal integrated patterns.

Relevant Section in the New Manuscript: Lines 447 – 470

The obtained 2D scattering images scans were transformed into one-dimensional (1D) scattering patterns using the azimuthal integration method in which the scattered light signals were averaged along the concentric circle around the incident beam by the wavevector (q) or the azimuth angle ($\psi = 360^\circ$).^{67,68} For the SAXS analysis, liquid neat DCPD and DCPD/AC mixtures were frontally polymerized in separate borosilicate capillary tubes ($d = 1.5$ mm) by

applying a localized thermal stimulus through the bottom of the tubes. Similar to our aforementioned results, more numbers of larger bubbles were formed with increasing AC wt% (Figure S11b-e). For a clearer assessment of the crystalline domain structure formed by different AC wt%, scattering light intensities obtained for the DCPD/0.5-4 wt% AC foams were background subtracted by that of neat DCPD. In the obtained azimuthal-integrated intensity plot, structure peaks pertaining to 0.5 wt%, 1 wt%, and 4 wt% AC were observed at approximately $\psi_{0.5\text{wt}\% \text{AC}} = 29^\circ, 161^\circ, 178^\circ$; $\psi_{1\text{wt}\% \text{AC}} = 29^\circ, 152^\circ, 173^\circ$; and $\psi_{4\text{wt}\% \text{AC}} = 75^\circ, 161^\circ, 173^\circ$ (Figure 6c). Unlike the scattering patterns of these formulations, that of 2wt% AC exhibited rather periodic pattern of several structural peaks between $\psi \sim 29^\circ$ and 132° , which may suggest a relative homogeneous distribution of crystalline domains along its vertical axis. This phenomenon could be supported by the scattering intensity vs. scattering vector ($\ln(I(q))$ vs. q) plot of 2wt% AC. While no intensity peaks were observed from other AC wt%, sharp intensity peaks at $q_{\text{max}} = 0.02846 \text{ \AA}^{-1}$ and $q_{\text{min}} = 0.03070 \text{ \AA}^{-1}$ were observed for 2wt% AC (Figure 6d). By the equation $d_L = 2\pi/q_{\text{max}}$ ^{68,69}, the crystalline domain spacing (d_L) of 2wt% AC was calculated as $d_L = 3.514 \text{ nm}$. Based on these results, it could be understood that a long-range order of crystalline domains was formed by 2wt% AC that were independent of those generated by DCPD.

Relevant Section in the New Manuscript: Line 350, Figure 6

Line 339: Figures S3a and S3c do not correspond to the current discussion.

We greatly appreciate the reviewer's detailed comment that pointed out the error in the original manuscript. With the newly prepared manuscript, the relevant section was adjusted appropriately.

Relevant Section in the New Manuscript: Lines 401 – 508

Line 339-347: Authors suggest that polydispersity is problematic in the determination of crystallinity via SAXS. However, they claim the samples are more crystalline which indicates a higher anisotropy. If SAXS generated unclear results about the alignment of the AC, the authors need to include other experiments that corroborate their assumption

We certainly agree with the reviewer's comment pointing out the lack of elaboration on the claims we had made in our original manuscript. Based on the results obtained from the ATR-FTIR, SAXS, and WAXS analyses, our group proposed a different interpretation regarding the oxidation resistance of the DCPD/2wt% AC foam and the long-range order of crystalline domains that was presumably induced by the introduction of 2wt% AC. The new findings were described in the relevant sections of the new manuscript.

Relevant Section in the New Manuscript: Line 447 – 470

The obtained 2D scattering images scans were transformed into one-dimensional (1D) scattering patterns using the azimuthal integration method in which the scattered light signals were averaged along the concentric circle around the incident beam by the wavevector (q) or the azimuth angle ($\psi = 360^\circ$).^{67,68} For the SAXS analysis, liquid neat DCPD and DCPD/AC mixtures were frontally polymerized in separate borosilicate capillary tubes ($d = 1.5$ mm) by applying a localized thermal stimulus through the bottom of the tubes. Similar to our aforementioned results, more numbers of larger bubbles were formed with increasing AC wt% (Figure S11b-e). For a clearer assessment of the crystalline domain structure formed by different AC wt%, scattering light intensities obtained for the DCPD/0.5-4 wt% AC foams were background subtracted by that of neat DCPD. In the obtained azimuthal-integrated intensity plot, structure peaks pertaining to 0.5 wt%, 1 wt%, and 4 wt% AC were observed at approximately $\psi_{0.5\text{wt}\% \text{AC}} = 29^\circ, 161^\circ, 178^\circ$; $\psi_{1\text{wt}\% \text{AC}} = 29^\circ, 152^\circ, 173^\circ$; and $\psi_{4\text{wt}\% \text{AC}} = 75^\circ, 161^\circ$,

173° (Figure 6c). Unlike the scattering patterns of these formulations, that of 2wt% AC exhibited rather periodic pattern of several structural peaks between $\psi \sim 29^\circ$ and 132° , which may suggest a relative homogeneous distribution of crystalline domains along its vertical axis. This phenomenon could be supported by the scattering intensity vs. scattering vector ($\ln(I(q))$ vs. q) plot of 2wt% AC. While no intensity peaks were observed from other AC wt%, sharp intensity peaks at $q_{\max} = 0.02846 \text{ \AA}^{-1}$ and $q_{\min} = 0.03070 \text{ \AA}^{-1}$ were observed for 2wt% AC (Figure 6d). By the equation $d_L = 2\pi/q_{\max}$ ^{68,69}, the crystalline domain spacing (d_L) of 2wt% AC was calculated as $d_L = 3.514 \text{ nm}$. Based on these results, it could be understood that a long-range order of crystalline domains was formed by 2wt% AC that were independent of those generated by DCPD. Then, the effects of introducing ACs to the crystal and lattice structures of DCPD were investigated through the WAXS analysis.

Line 346-349: Authors need to expand on the relation between DCPD oxidation and the foaming mechanism.

We are grateful for the reviewer's comment on the initially proposed mechanism. After conducting additional experiments based on the reviewer's suggestion, we present further evidence that may support the proposed mechanism of relative oxidation resistance observed from the DCPD/2wt% AC foam.

As shown in Figure 6a, the relative oxidation resistance of DCPD/2wt% AC was confirmed from the ATR-FTIR analysis of neat DCPD and the produced DCPD/AC foams. It was previously studied that the proneness of the DCPD alkene backbone towards oxidative damage was the cause of DCPD foam oxidation.⁶² Therefore, hydrogenation of the polymerized DCPD material was mainly highlighted as a method to prevent oxidation for its ability to affect the C=C chain conformation.^{63,64} Since our experimental procedure was devoid of hydrogenation posttreatment procedure, we sought to investigate the crystal structure of the formed DCPD/2wt% AC foam.

First, the 2D small-angle X-ray scattering (SAXS) analysis was performed for neat DCPD solid and DCPD/0.5 – 4wt% AC foams by frontally polymerizing the specimens in 1.5 mm borosilicate capillary tubes. The SAXS analysis was conducted in a transmission mode where the primary beam was emitted in a perpendicular direction to the vertical axes of the samples (Figure 6b). Here, the collected scattering pattern of neat DCPD was considered the background. Therefore, the analyses of background subtracted DCPD/AC patterns provided the information about the crystalline domains that were formed by the varying amounts of AC. From the azimuthal integrated SAXS patterns of ACs (Figure 6c), a more homogenous distribution of crystalline domains was observed across $\psi \sim 29^\circ$ and 132° for 2wt% AC, while other AC wt% displayed rather heterogenous distribution of structure peaks. In Fig. R1, we show how our group interpreted the obtained 1D SAXS patterns of the varying AC wt%. Moreover, from the $(\ln(I(q)))$ vs q plot of 2wt% AC, a long-range order of crystalline domains was confirmed by the sharp peaks at $q_{\max} = 0.02846 \text{ \AA}^{-1}$ and $q_{\min} = 0.03070 \text{ \AA}^{-1}$ (Figure 6d). By the equation $d_L = 2\pi/q_{\max}^{68,69}$, the crystalline domain spacing (d_L) of 2wt% AC was calculated as $d_L = 3.514 \text{ nm}$. Then, the effects of incorporating ACs to the crystal structure of frontally polymerized DCPD was studied by the wide-angle X-ray scattering analysis (WAXS). For this technique, the formed neat DCPD solid and DCPD/AC foams were cut into rectangular shards with approximately 3 mm thickness. As shown in Fig. 6e and 6f, a broad shoulder peak was observed from the DCPD/2wt% AC sample which suggested the generated orderedness of the DCPD crystal structure by the addition of 2wt% AC. This phenomenon may be supported by the general trend of 2θ shifts with increasing AC wt%, as shown in the intensity vs. scattering angle (2θ) plot (Figure 6g and 6h). Especially, the largest 2θ shift was observed for the DCPD/2wt% AC sample ($2\theta_{2\text{wt}\% \text{ AC}} = 16.88^\circ$). Applying this value to the Bragg's- ($\lambda = 2d\cos\theta$) and the Scherrer ($D = (K\lambda)/(\beta\cos\theta)$) equations, the lattice spacing (d-spacing) and crystallite size (D) pertaining to the DCPD/2wt% AC foam was calculated.^{64,66,68} As shown in Table 4,

d-spacing_{2wt% AC} = 0.5248 nm and $D_{2wt\% AC}$ = 18.19 nm were obtained. Based on these findings, it could be understood that the crystal structure of DCPD/2wt% AC was a neat organization of tightly packed crystallites that relatively suppressed oxygen diffusion of the produced foam.^{61,62,74} Therefore, a one-pot strategy of enhancing oxidation resistance of pDCPD foams by the incorporation of an optimal amount of 2wt% AC is presented in this study.

Figure. R1 A schematic showing the relative homogeneous distribution of crystalline domains that was detected in the SAXS analysis of 2wt% AC.

Relevant Section in the New Manuscript: Lines 401 – 508

Relevant Section in the New Manuscript: Line 350, Figure 6

Figure 6. (a) ATR-FTIR spectra of the prepared neat DCPD and DCPD/AC foams are shown. Compared to the spectra obtained shortly after the foams were produced, those taken after 8 weeks displayed significantly oxidated states except for DCPD/2wt% AC. (b) A schematic of a transmission mode SAXS and WAXS analysis. (c) The background subtracted azimuthal integrated SAXS pattern of 2wt% AC shows a rather homogeneous distribution of crystalline domains at $\psi \sim 29^\circ$ and 132° . (d) A long-range order of crystalline domains induced by 2wt% AC is shown by the broad shoulder peak. (e) – (f) WAXS patterns of neat DCPD and DCPD/AC foams show 2wt% AC may have induced directionality in the lattice structures. (g) – (h) The effects of incorporating ACs to the frontally polymerized DCPD crystal structures are observed by the shifts in 2θ , while DCPD/2wt% AC exhibit the lowest d-spacing and the smallest crystallite size.

Relevant Section in the New Manuscript: **Table 4**

Table 4. The shifts in 2θ values, lattice spacing (d-spacing), and crystallite sizes (D) were determined by the WAXS analyses of neat DCPD and DCPD/0.5 – 4wt% AC foams.

	0 wt%	0.5 wt%	1 wt%	2 wt%	4 wt%
2θ (deg)	16.36	16.52	16.58	16.88	16.76
Lattice Spacing (d-spacing) (nm)	0.5414	0.5362	0.5342	0.5248	0.5286
Crystallite Size (D)	23.48	20.90	20.47	18.21	19.38

Line 353: It is hard to see the difference in oxidation with the images provided in Figure 7b. FTIR characterization will be helpful to corroborate their differences.

We appreciate the reviewer’s advice to provide more quantitative data that show oxidation of DCPD solids over time. Here we present a plot showing the attenuated total reflectance-fourier-transform infrared spectroscopy (ATR-FTIR) spectra of the prepared neat DCPD and DCPD/AC samples. The FTIR spectra were measured by putting the probes in contact with both the outer and the inner layers of the cellular solids which resulted similar spectrum patterns. Compared to those obtained after being freshly made, the FTIR spectra of the prepared DCPD solids attained after 8 weeks revealed their oxidated states which were supported by the apparent C=O and O-H absorptions at 1700 cm^{-1} and 3390 cm^{-1} , respectively, that agreed with the previous work reported by S. Kovačič *et al.*¹.

Relevant Section in the New Manuscript: Line 416 – 428, Figure 6a

Firstly, we observed discoloration of the produced DCPD/AC foams to brown over the span of 8 weeks post-production as compared to the yellowish-white color they had after being freshly made (Figure S14). While being described as a physical sign of materials degradation in ambience, DCPD oxidation was previously highlighted by S. Kovačič *et al.*^{36,62} as a major challenge in the production of porous DCPD monoliths and membranes as it is often followed by the drastic alterations to materials properties and reduced product applicability. Experimentally, oxidation of DCPD/AC foams was clearly observed from the normalized ATR-FTIR spectra of the prepared samples that displayed pronounced C=O and O-H absorption peaks at around 1700 cm^{-1} and 3390 cm^{-1} , respectively, which agreed with the previously reported results (Figure 6a).⁶² Unlike those of the other DCPD/AC formulations, however, the IR spectrum of DCPD/2wt% AC exhibited significantly lower intensity of the C=O and O-H peaks, that confirmed the relative oxidation-resisting ability of DCPD/2wt% AC foam.

Line 369-372: How do the authors know that an organization of the polymerizing DCPD is happening in their system? It is hard to support this claim just by referencing literature on other systems, which are dramatically different from the current one.

We greatly appreciate the reviewer's concern over the reliability of the previous claim we made in the original manuscript. To support the newly proposed relationship between oxidation of DCPD/AC foams and the formed crystal structures, we subtracted the obtained SAXS intensity of neat DCPD from those obtained from the DCPD/AC foams. Therefore, by presenting background subtracted SAXS patterns of ACs, we have shown the formation of crystalline domains that was purely induced by ACs.

Relevant Section in the New Manuscript: Lines 401 – 508

Relevant Section in the New Manuscript: Figure 6c, 6d

Line 372-375: Authors need to provide quantitative oxidation and crystallinity values for all the samples to support that they are correlated with each other.

We greatly appreciate the reviewer's advise to present quantitative evidence that our observation for oxidation may be explained. With the newly presented finds that were elaborated in pages 22 – 26, we highlight the calculated crystalline domain spacing ($d_L = 3.514$ nm), lattice spacing (d-spacing) and crystallite size (D) values that were obtained from the SAXS and WAXS analyses.

Relevant Section in the New Manuscript: Lines 401 – 508

Relevant Section in the New Manuscript: Table 4

Table 4. The shifts in 2θ values, lattice spacing (d-spacing), and crystallite sizes (D) were determined by the WAXS analyses of neat DCPD and DCPD/0.5 – 4wt% AC foams.

	0 wt%	0.5 wt%	1 wt%	2 wt%	4 wt%
2θ (deg)	16.36	16.52	16.58	16.88	16.76
Lattice Spacing (d-spacing) (nm)	0.5414	0.5362	0.5342	0.5248	0.5286
Crystallite Size (D)	23.48	20.90	20.47	18.21	19.38

Line 383: How do the authors know that the 2wt% mixture has a stoichiometric amount of functional C=C and chiral-inducing hydroxyl groups?

We appreciate the reviewer’s insightful comment pointing out the claim that lacked sufficient corroboration from our original manuscript. Based on the new findings we have made through the reviewer’s thoughtful comments and suggestions, the newly edited manuscript described the effect of 2wt% AC to have induced a long-range order of crystalline domains that was independent of the crystalline domains that may have formed by DCPD. Moreover, from the WAXS patterns, we showed that the introduction of 2wt% AC to the crystal structure of DCPD resulted the narrowest lattice spacing and thus the smallest crystallite sizes. Collectively, these results supported the relative oxidation resistance that was observed from the DCPD/2wt% foam.

Relevant Section in the New Manuscript: Lines 401 – 508

Line 385-387: The authors use the word “synergistically” to explain their claims about the organization of the CN. However, a better discussion is needed to support this hypothesis.

Our group appreciate the reviewer’s concern over the validity in the claims we had made in the original manuscript. Based on the new findings, we have made appropriate changes to the relevant section of the new manuscript.

Relevant Section in the New Manuscript: Lines 401 – 508

Line 400-418: Conclusions need to be rewritten based on the suggestion made along with the manuscript revision.

We are grateful for the reviewer’s insightful suggestions that we are ascertained of their importance to significantly raising the quality of the current study. Based on all the corrections made in the new manuscript, we have made changes in our conclusions.

Relevant Section in the New Manuscript: Conclusions

In summary, we have demonstrated an efficient method of producing solid polymeric foams by frontally polymerizing DCPD/AC mixture. Two allyl-functionalized CNC samples were prepared by the varying % allylation: 79 % (AC) and 13 % (LAC). Both AC and LAC were dispersed in liquid monomer at different wt%: 0.5, 1, 2, 4. FP of DCPD/AC and -LAC mixtures were initiated by a localized thermal stimulus and cellular solids with varying pore microstructures were produced by controlling the AC and LAC wt%. μ -CT scan analyses clearly demonstrated that for increasing % allylation and wt% of allylated CNCs, the pore size and distribution became larger and increasingly homogeneous, respectively. Moreover, with increasing AC wt%, % porosity of the foams increased while maintaining the relative DA. The FE-SEM analysis and the obtained v_f , T_{Max} , and $t_{initiation}$ values displayed the effects of varying crosslinking degree to the construct of the formed DCPD/AC foams by the amount of C=C

moiety on ACs. Amongst the produced DCPD/AC foams, the relative oxidation resistance of DCPD/2wt% AC foam was highlighted by the ATR-FTIR spectra. The background subtracted SAXS pattern of 2wt% AC exhibited a formation of long-range order of the crystalline domains that was independent from those that may have formed by DCPD. From the WAXS patterns of DCPD/AC foams, the immediate effects of introducing ACs to the crystal structures of DCPD was observed. Meanwhile, with the incorporation of 2wt% AC, the formed DCPD/AC crystal structure displayed the smallest lattice spacing and the subsequent crystallite size. In conclusion, we have demonstrated that an optimal amount of 2wt% AC not only have directed the formation of honeycomb-like channels but also induced a formation of the more neatly packed crystal structure and produced DCPD/2wt% AC foam with a relative oxidation resistance.

[Response to REVIEWER #3]

Dear Reviewer #3,

We would like to express our greatest appreciation for your considerate comments towards our manuscript. Our group believe that a significant improvement to the quality of our work was made by your questions and suggestions.

Once again, we are honored to have received these important points from the reviewer and we would love to express our immense respect for your time and consideration.

Thank you so much.

Sincerely,

Prof. Seung-Yeop Kwak

The authors state that “For increasing AC wt%, paler and more porous foams were produced (Figure 3a), but it is not obvious from the figure. High quality image should be provided.

We greatly appreciate the reviewer’s concern over the low visibility of the images showing the produced DCPD/AC foams. Here we provide images of higher quality and greater magnification to a better view of the differences in color and pore sizes between the prepared cellular solids.

Relevant Section in the New Manuscript: **Figure 1**

In FP process, the temperature profile is highlighted with a stable horizontal line and a maximum point. This temperature profile should be tested at an unreacted fixed point. Thus, the Figure 6c might be retested.

We agree with the reviewer that an unreacted temperature profile should be included in the temperature profile of FP. Moreover, our group concluded that the previous method of measuring reaction temperature with an infrared camera through the glass tube did not provide the most accurate temperature values. Therefore, we obtained a new FP temperature profile using a T-type thermocouples that were purchased from Omega Engineering which were submerged in the liquid monomer/AC mixture before the initiation of reaction, and recorded temperature values every two seconds.

The new temperature ($^{\circ}\text{C}$) vs time (s) plot, which contains measured temperature values of neat DCPD and DCPD/AC 0.5, 1, 2, 4 wt%, was characterized by sharp temperature spikes i.e., T_{Max} , that were observed shortly after the reaction was initiated while at an unreacted state, temperature stayed constant at around 21°C . On the other hand, the T_{Max} values of the prepared specimen displayed little differences at around 151°C . previous works have shown that FP systems containing NPs exhibited a gradual decrease in T_{Max} with increasing NP wt% due to their ability of absorbing heat released from the reaction.^{1,2,3} Our group reduced the size of ACs as a leading factor to induce this effect because compared to the NP size ($w \times l$) (14 x 88 nm) in this study fell in between those of spherical SiO_2 “heatsink” NPs that ranged from 20 nm to 10 nm that were reported by S. Chen *et al*¹ and S. P. Davtyan *et al*², respectively. Meanwhile, L. M. Dean *et al*⁴ recently reported a work attributing enhancement in thermal conductivity during FP to the high aspect ratio of carbon nanotubes (CNTs). This group also highlighted the intrinsic photothermal conversion efficiency found in CNTs as another factor of improved T_{Max} values during FP.⁴ Based on these previous studies, the phenomenon presented in our study can be explained by taking in account for the high aspect ratio and

nonconductivity of ACs. Our group postulate that these two characteristics offset each other; the anisotropy found in AC morphology enhanced energy transfer which was partially lost due to the nonconducting nature of AC NP thus resulting in the small T_{Max} shifts compared to that of neat DCPD.

As for the rather increasing standard deviation of T_{Max} values, we presume that the stochastic formation of bubbles during FP of DCPD/AC made it challenging for the thermocouple sensor to not only detect the sole instantaneous T_{Max} of the dynamic pDCPD front but also to differentiate the temperature of pDCPD and the simultaneously generated air-filled voids (Figure S13a). This phenomenon may be further demonstrated from the μ -CT images that visualized the positions of the thermocouple hot junctions being inside the pores of the DCPD/AC foams (Figure S13b-e).

These findings and the related explanations have been edited into relevant sections in the new manuscript.

References

1. Chen, S., Sui, J., Chen, L. & Pojman J. A. Polyurethane-Nanosilica Hybrid Nanocomposites Synthesized by Frontal Polymerization. *J. Polym. Sci. A Polym. Chem.*, **43**, 1670-1680 (2005).
2. Davtyan, S. P., Berlin, A. A., Shik, K., Tonoyan, A. O. & Rogovina, S. Z. Polymer Nanocomposites with a Uniform Distribution of Nanoparticles in a Polymer Matrix Synthesized by Frontal Polymerization Technique. *Nanotechnol Russia*. **4**, 489-498 (2009).
3. Gao, Y. *et al.* Controllable Frontal polymerization and spontaneous patterning enabled by phase-changing particles. *Small* **17**, 2102217 (2021).
4. Dean, L. M., Ravindra, A., Guo, A. X., Yourdkhani, M. & Sottos, N. R. Photothermal initiation of frontal polymerization using carbon nanoparticles. *ACS Appl. Polym. Mater.* **2**,

Relevant Section in the New Manuscript: **Figure 5c**

Figure 6. (c) The temperature vs. time plots exhibited negligible changes in T_{Max} for increasing AC wt%. On the other hand, a general trend of increasing reaction initiation time ($t_{initiation}$) was observed with increasing AC wt%.

Relevant Section in the New Manuscript: **Table 3**

Table 3. Front velocity (v_f), maximum temperature (T_{Max}), and initiation time ($t_{initiatoin}$) values were obtained during frontal polymerization of the DCPD/AC samples.

	0 wt%	0.5 wt%	1 wt%	2 wt%	4 wt%
V_f (mm s ⁻¹)	0.89 ± 0.2	1.2 ± 0.05	1.3 ± 0.03	1.7 ± 0.1	1.1 ± 0.1
T_{Max} (°C)	151 ± 0.20	153 ± 5.4	150 ± 9.3	153 ± 17	151 ± 15
$t_{initiation}$ (s)	22 ± 6	60 ± 1	106 ± 4	36 ± 8	110 ± 8

Relevant Section in the New Manuscript: **Line 291**

Fig. 6c shows a new temperature ($^{\circ}\text{C}$) vs time (s) plot in which temperature peaks i.e., maximum temperature (T_{Max}), were observed from the prepared specimens. The flatter region from each plot represents contact time of the soldering iron to the glass tube before foaming processes were observed. Without thermal initiation, the temperature of the DCPD/AC mixture remained constant at approximately 22°C . Upon the application of thermal stimulus, neat DCPD and DCPD/AC 0.5-4wt% samples displayed minimal differences in their T_{Max} values at around 151°C which falls within the previously reported region of T_{Max} values observed during FP of DCPD.²⁴ On the other hand, earlier works reported that for increasing the wt% of micro-/nanoparticles, T_{Max} decreased during FP of DCPD due to the ability of the filler to absorb heat released from the reaction.^{21, 47}

Relevant Section in the New Manuscript: Line 458-460

T-type welded thermocouples were purchased from Omega Engineering and were used as received. Thermocouples were submerged to the bottom of the glass tubes from where thermal stimulus was applied and the reaction of FP was initiated.

Relevant Section in the New Manuscript: **Figure S13**

Figure S13. (a) A schematic is shown in support of our presumption that the stochastically growing nature of bubbles led to the large standard deviation due to the incomplete contact between the thermocouple hot junction and the polymerizing monomer. (b) μ -CT images of the polymerized foams showed that the hot junctions of the thermocouples were sometimes located inside the pores (dark areas) or in contact with the polymer (grey area).

The authors state that “the more obvious sign of DCPD oxidation in air is the color change from white to brown. This is not rigorous. Some quantitative analysis should be conducted.

We appreciate the reviewer’s advice to provide more quantitative data that show oxidation of DCPD solids over time. Here we present a plot showing the attenuated total reflectance-fourier-transform infrared spectroscopy (ATR-FTIR) spectra of the prepared neat DCPD and DCPD/AC samples. The FTIR spectra were measured by putting the probes in contact with both the outer and the inner layers of the cellular solids which resulted similar spectrum patterns. Compared to those obtained after being freshly made, the FTIR spectra of the prepared DCPD solids attained after 8 weeks revealed their oxidated states which were supported by the apparent C=O and O-H absorptions at 1700 cm^{-1} and 3390 cm^{-1} , respectively, that agreed with the previous work reported by S. Kovačič *et al.*¹.

Relevant Section in the New Manuscript: Line 416 – 428, Figure 6a

Firstly, we observed discoloration of the produced DCPD/AC foams to brown over the span of 8 weeks post-production as compared to the yellowish-white color they had after being freshly made (Figure S14). While being described as a physical sign of materials degradation in ambience, DCPD oxidation was previously highlighted by S. Kovačič *et al.*^{36,62} as a major challenge in the production of porous DCPD monoliths and membranes as it is often followed by the drastic alterations to materials properties and reduced product applicability. Experimentally, oxidation of DCPD/AC foams was clearly observed from the normalized ATR-FTIR spectra of the prepared samples that displayed pronounced C=O and O-H absorption peaks at around 1700 cm^{-1} and 3390 cm^{-1} , respectively, which agreed with the previously reported results (Figure 6a).⁶² Unlike those of the other DCPD/AC formulations, however, the

IR spectrum of DCPD/2wt% AC exhibited significantly lower intensity of the C=O and O-H peaks, that confirmed the relative oxidation-resisting ability of DCPD/2wt% AC foam.

The proposed mechanism needs to be further elucidated.

We are grateful for the reviewer's comment on the initially proposed mechanism. After conducting additional experiments based on the reviewer's suggestion, we present further evidence that may support the proposed mechanism of relative oxidation resistance observed from the DCPD/2wt% AC foam.

As shown in Figure 6a, the relative oxidation resistance of DCPD/2wt% AC was confirmed from the ATR-FTIR analysis of neat DCPD and the produced DCPD/AC foams. It was previously studied that the proneness of the DCPD alkene backbone towards oxidative damage was the cause of DCPD foam oxidation.⁶² Therefore, hydrogenation of the polymerized DCPD material was mainly highlighted as a method to prevent oxidation for its

ability to affect the C=C chain conformation.^{63,64} Since our experimental procedure was devoid of hydrogenation posttreatment procedure, we sought to investigate the crystal structure of the formed DCPD/2wt% AC foam.

First, the 2D small-angle X-ray scattering (SAXS) analysis was performed for neat DCPD solid and DCPD/0.5 – 4wt% AC foams by frontally polymerizing the specimens in 1.5 mm borosilicate capillary tubes. The SAXS analysis was conducted in a transmission mode where the primary beam was emitted in a perpendicular direction to the vertical axes of the samples (Figure 6b). Here, the collected scattering pattern of neat DCPD was considered the background. Therefore, the analyses of background subtracted DCPD/AC patterns provided the information about the crystalline domains that were formed by the varying amounts of AC. From the azimuthal integrated SAXS patterns of ACs (Figure 6c), a more homogenous distribution of crystalline domains was observed across $\psi \sim 29^\circ$ and 132° for 2wt% AC, while other AC wt% displayed rather heterogenous distribution of structure peaks. In Fig. R1, we show how our group interpreted the obtained 1D SAXS patterns of the varying AC wt%. Moreover, from the $(\ln(I(q))$ vs q) plot of 2wt% AC, a long-range order of crystalline domains was confirmed by the sharp peaks at $q_{\max} = 0.02846 \text{ \AA}^{-1}$ and $q_{\min} = 0.03070 \text{ \AA}^{-1}$ (Figure 6d). By the equation $d_L = 2\pi/q_{\max}$ ^{68,69}, the crystalline domain spacing (d_L) of 2wt% AC was calculated as $d_L = 3.514 \text{ nm}$. Then, the effects of incorporating ACs to the crystal structure of frontally polymerized DCPD was studied by the wide-angle X-ray scattering analysis (WAXS). For this technique, the formed neat DCPD solid and DCPD/AC foams were cut into rectangular shards with approximately 3 mm thickness. As shown in Fig. 6e and 6f, a broad shoulder peak was observed from the DCPD/2wt% AC sample which suggested the generated orderedness of the DCPD crystal structure by the addition of 2wt% AC. This phenomenon may be supported by the general trend of 2θ shifts with increasing AC wt%, as shown in the intensity vs. scattering angle (2θ) plot (Figure 6g and 6h). Especially, the largest 2θ shift was observed for the

DCPD/2wt% AC sample ($2\theta_{2\text{wt}\% \text{ AC}} = 16.88^\circ$). Applying this value to the Bragg's- ($\lambda = 2d\cos\theta$) and the Scherrer ($D = (K\lambda)/(\beta\cos\theta)$) equations, the lattice spacing (d-spacing) and crystallite size (D) pertaining to the DCPD/2wt% AC foam was calculated. ^{64,66,68} As shown in Table 4, $d\text{-spacing}_{2\text{wt}\% \text{ AC}} = 0.5248 \text{ nm}$ and $D_{2\text{wt}\% \text{ AC}} = 18.19 \text{ nm}$ were obtained. Based on these findings, it could be understood that the crystal structure of DCPD/2wt% AC was a neat organization of tightly packed crystallites that relatively suppressed oxygen diffusion of the produced foam.^{61,62,74} Therefore, a one-pot strategy of enhancing oxidation resistance of pDCPD foams by the incorporation of an optimal amount of 2wt% AC is presented in this study.

Figure. R1 A schematic showing the relative homogeneous distribution of crystalline domains that was detected in the SAXS analysis of 2wt% AC.

Relevant Section in the New Manuscript: Line 350, Figure 6

Figure 6. (a) ATR-FTIR spectra of the prepared neat DCPD and DCPD/AC foams are shown. Compared to the spectra obtained shortly after the foams were produced, those taken after 8 weeks displayed significantly oxidated states except for DCPD/2wt% AC. (b) A schematic of a transmission mode SAXS and WAXS analysis. (c) The background subtracted azimuthal integrated SAXS pattern of 2wt% AC shows a rather homogeneous distribution of crystalline domains at $\psi \sim 29^\circ$ and 132° . (d) A long-range order of crystalline domains induced by 2wt% AC is shown by the broad shoulder peak. (e) – (f) WAXS patterns of neat DCPD and DCPD/AC foams show 2wt% AC may have induced directionality in the lattice structures. (g) – (h) The effects of incorporating ACs to the frontally polymerized DCPD crystal structures are observed by the shifts in 2θ , while DCPD/2wt% AC exhibit the lowest d-spacing and the smallest crystallite size.

Relevant Section in the New Manuscript: Table 4

Table 4. The shifts in 2θ values, lattice spacing (d-spacing), and crystallite sizes (D) were determined by the WAXS analyses of neat DCPD and DCPD/0.5 – 4wt% AC foams.

	0 wt%	0.5 wt%	1 wt%	2 wt%	4 wt%
2θ (deg)	16.36	16.52	16.58	16.88	16.76
Lattice Spacing (d-spacing) (nm)	0.5414	0.5362	0.5342	0.5248	0.5286
Crystallite Size (D)	23.48	20.90	20.47	18.21	19.38

Reviewer #1 (Remarks to the Author):

The authors followed the remarks of the reviewers and did a lot of experimental work to improve their manuscript. Every single point was explained in great detail. In my opinion this manuscript is now ready to be published.

Reviewer #3 (Remarks to the Author):

After revision, the authors did quite a few works to improve the quality of paper. I would recommend this manuscript for this format to be accepted and published.